# Stay Hungry, Keep Learning: Sustainable Plasticity for Deep Reinforcement Learning

Huaicheng Zhou [1]  Zifeng Zhuang [1]  Donglin Wang [1]

## Abstract

The integration of Deep Neural Networks in Reinforcement Learning (RL) systems has led to remarkable progress in solving complex tasks but also introduced challenges like primacy bias and dead neurons. Primacy bias skews learning towards early experiences, while dead neurons diminish the network's capacity to acquire new knowledge. Traditional reset mechanisms aimed at addressing these issues often involve maintaining large replay buffers to train new networks or selectively resetting subsets of neurons. However, these approaches either incur prohibitive computational costs or reset network parameters without ensuring stability through recovery mechanisms, ultimately impairing learning efficiency. In this work, we introduce the novel concept of neuron regeneration, which combines reset mechanisms with knowledge recovery techniques. We also propose a new framework called Sustainable Backup Propagation(SBP) that effectively maintains plasticity in neural networks through this neuron regeneration process. The SBP framework achieves whole network neuron regeneration through two key procedures: cycle reset and inner distillation. Cycle reset involves a scheduled renewal of neurons, while inner distillation functions as a knowledge recovery mechanism at the neuron level. To validate our framework, we integrate SBP with Proximal Policy Optimization (PPO) and propose a novel distillation function for inner distillation. This integration results in Plastic PPO (P3O), a new algorithm that enables efficient cyclic regeneration of all neurons in the actor network. Extensive experiments demonstrate the approach effectively maintains policy plasticity and improves sample efficiency in reinforcement learning.

[1]School of Engineering, Westlake University, Hangzhou, China. Correspondence to: Donglin Wang <wang-donglin@westlake.edu.cn>.

*Proceedings of the $42^{nd}$ International Conference on Machine Learning*, Vancouver, Canada. PMLR 267, 2025. Copyright 2025 by the author(s).

## 1. Introduction

Deep reinforcement learning has advanced significantly through the integration of deep neural networks, resulting in notable achievements across various domains (Singh et al., 2022; Arulkumaran et al., 2017; Yu et al., 2021). Despite these advancements, a critical issue that has emerged is the loss of plasticity, as detailed in (Lyle et al., 2023; Abbas et al., 2023). This refers to the diminishing ability of a network to learn and adapt over time. As network neurons become saturated, they "become full", losing the capacity to incorporate new information effectively. This reduction in plasticity primarily affects the neurons in the network, leading to decreased effectiveness and eventually causing neurons to become dead (Lu et al., 2019; Shin & Karniadakis, 2020) or dormant (Sokar et al., 2023b). Additionally, the problem of overfitting in deep learning, known as primacy bias (Nikishin et al., 2022), further causes this loss of plasticity. Consequently, there is an urgent imperative to develop mechanisms for the repair or revitalization of neurons affected by primacy bias or those that have lapsed into dormancy, with the objective of reawakening their "hunger" for novel information.

Reset mechanisms have been proven to be effective measures for addressing the loss of plasticity in neural networks. However, existing reset approaches have demonstrated various limitations. Early studies (Nikishin et al., 2022; D'Oro et al., 2022; Kim et al., 2024) proposed resetting either the final layer or all neurons to revitalize learning capabilities, but these methods often led to a performance-resource trade-off, requiring additional training to recover lost performance. More targeted approaches, such as CBP (Dohare et al., 2021) and ReDo (Sokar et al., 2023b), focused on selectively resetting non-contributing neurons. While this strategy reduced information loss, it only partially restored plasticity and achieved limited performance improvements. A fundamental challenge across these methods is that neurons reaching critical importance become either irresetable or damaging when reset, creating a dilemma that renders the reset strategy ineffective and potentially limiting the network's overall potential. This limitation underscores the need for a more sophisticated approach that can enhance network plasticity while maintaining critical knowledge.

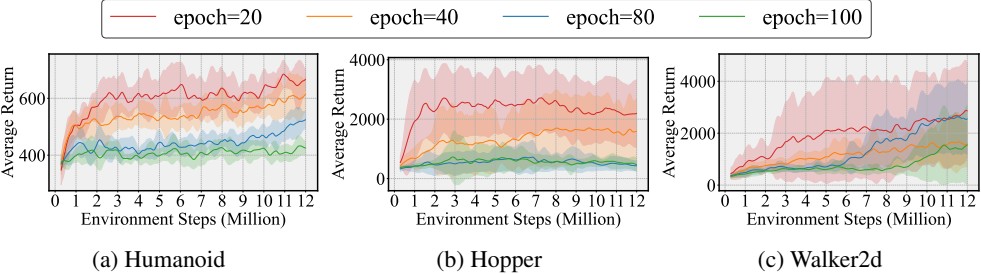

*Figure 1.* Performance of PPO across varying numbers of training epochs of actor network per batch. Increasing training epochs impedes performance improvement.

Inspired by regenerative processes in human cells (Carlson, 2011), we propose the concept of *neuron regeneration*, a biomimetic approach that aims to recover plasticity while preserving crucial knowledge within the network. Similar to how biological cells maintain their vitality through regeneration, our *neuron regeneration* mechanism enables the renewal of neural network components while retaining essential learned information. This process enables neural networks to maintain long-term learning capabilities without performance degradation. To implement *neuron regeneration*, we introduce the Sustainable Backup Propagation (SBP) approach, which integrates reset and distillation mechanisms into traditional backpropagation(Hecht-Nielsen, 1992). Drawing inspiration from natural cellular regeneration cycles(Sender & Milo, 2021), SBP employs a cyclic reset strategy to mitigate neuron plasticity degradation and primacy bias. The key innovation lies in our *inner distillation* process, which facilitates knowledge transfer from reset neurons to others, ensuring effective regeneration without performance loss. This approach not only maintains the network's capacity to absorb new information but also perpetuates its learning process, maximizing its potential for continuous growth.

As shown in Figure 1, Proximal Policy Optimization (PPO) (Schulman et al., 2017) exhibits significant primacy bias, where early training experiences disproportionately influence the final policy, leading to performance degradation as training progresses. This degradation stems from the loss of neuron plasticity, limiting the network's ability to adapt to new experiences. To address this limitation, we integrated our SBP approach with PPO by incorporating neuron regeneration into the policy network. We developed a novel $\alpha$-weighted Double KL divergence ($\alpha$-DKL) loss function that dynamically balances knowledge preservation and update flexibility. This loss function employs two KL terms with an weighting mechanism to effectively filter out harmful information while retaining valuable policy knowledge during the regeneration process. The resulting Plastic PPO (P3O) algorithm implements neuron regeneration within the actor network to maintain learning efficiency and achieve sustainable plasticity. Extensive experiments

were conducted across diverse environments, including MuJoCo (Todorov et al., 2012), DeepMind Control Suite (Tassa et al., 2018), and a specially designed MuJoCo variant called Cycle Friction that tests adaptation to changing dynamics. Results demonstrate that P3O consistently outperforms standard PPO, achieving both higher average returns and more stable learning curves, validating the effectiveness of our neuron regeneration mechanism in maintaining policy plasticity while enhancing sample efficiency. Our contributions in this work can be summarized as follows:

- **Neuron Regeneration:** We introduce the concept of *neuron regeneration*, a biomimetic approach inspired by cellular regeneration processes. This novel mechanism maintains neural network plasticity while preserving learned knowledge, enabling continuous learning without performance degradation.
- **Sustainable Backup Propagation (SBP):** We propose SBP, a systematic framework that implements neuron regeneration through cyclic reset strategies and inner distillation mechanisms. By effectively addressing dead neurons and primacy bias, SBP ensures sustainable plasticity throughout the network's lifecycle.
- **Plastic PPO (P3O):** We introduce P3O, an enhanced version of PPO that integrates SBP and a novel $\alpha$-weighted Double KL divergence ($\alpha$-DKL) loss function. P3O overcomes the primacy bias problem in standard PPO, maintaining plasticity and improved sample efficiency across various reinforcement learning tasks.

## 2. Preliminaries and Related Work

### 2.1. On-policy Reinforcement Learning

Reinforcement learning is formalized as a Markov decision process (MDP) (Puterman, 2014). An MDP consists of a tuple $\langle S, A, R, P, \gamma \rangle$, where $S$ denotes the set of states, $A$ the set of actions, $R : A \times S \to \mathbb{R}$ a reward function, $P : S \times A \to P(S)$ a possibly stochastic transition probability function, $\gamma \in [0, 1)$ the discount factor. In reinforcement learning, the goal is to seek an optimal policy $\pi^* : S \to P(A)$ which maximizes the expected accumulated returns with discounted.

In on-policy reinforcement learning, the Proximal Policy Optimization (PPO) algorithm (Schulman et al., 2017) is utilized to update policies during interaction with the environment. The core objective function of PPO, denoted as $\mathcal{L}_{\text{clip}}(\theta)$, includes a clipping operation that sets gradients to zero when the probability ratio $r_t(\theta)$ falls outside $[1-\epsilon, 1+\epsilon]$. This prevents the policy from learning from advantages that would push it further outside the trust region, thereby enforcing trust region constraints in a computationally efficient manner. This operation ensures that the policy does not deviate excessively from the previous policy. The objective function is expressed as:

$$\mathcal{L}_{\text{clip}}(\theta) = \mathbb{E}_t[\min(r_t(\theta) \cdot \hat{A}_t, \text{clip}(r_t(\theta), 1-\epsilon, 1+\epsilon) \cdot \hat{A}_t)] \tag{1}$$

Here, $r_t(\theta) = \frac{\pi_\theta(a_t|s_t)}{\pi_{\theta_{\text{old}}}(a_t|s_t)}$ represents the probability ratio between new and old policies, and $\hat{A}_t$ is the estimated advantage function. The advantage function measures the value of the current policy relative to a baseline policy, calculated from trajectory data acquired during interactions with the environment. In on-policy training environments with dynamic data, where both input data and target values are nonstationary, network plasticity may be compromised, leading to suboptimal performance.

## 2.2. Plasticity in Reinforcement Learning

Plasticity, the ability of neural networks to adapt to new information, gradually declines over time, posing a critical challenge. This issue has prompted extensive research(Lyle et al., 2023; Abbas et al., 2023; Nikishin et al., 2022; Nauman et al., 2024; Dohare et al., 2024; Juliani & Ash, 2024; Lewandowski et al.) into methods for preserving neural network plasticity, with two main categories of approaches emerging to address this problem. The first involves various training techniques such as regularization, adjustments in activation functions, weight decay, and normalization strategies (Kumar et al., 2023; Delfosse et al., 2021; Lee et al., 2024a; Lyle et al., 2024; Elsayed et al., 2024). These methods delay plasticity loss by reducing overfitting and preventing large parameters.

The second category of approaches involves resetting the network (Nikishin et al., 2022; Schwarzer et al., 2023; Nikishin et al., 2024; Lee et al., 2024b) to recover the plasticity. This reset-based methodology addresses both neuron dormancy and primacy bias. Typically implemented by reinitializing weights of specific layers or the entire network, resets have been shown to effectively scale replay ratios, contributing to performance improvements (Kim et al., 2024; Xu et al., 2023). However, while resets can revive learning capabilities, they might lead to temporary performance degradation and require additional training to restore previously learned information. To mitigate these drawbacks, methods like CBP (Dohare et al., 2021) and ReDo (Sokar et al., 2023b) selectively reinitialize neurons deemed less useful based on certain metrics, minimizing the impact on overall performance. This approach highlights the delicate balance between recovering plasticity and maintaining network efficiency. However, there remains a need for more sophisticated methods that can effectively regenerate neurons while preserving learned knowledge, which is the focus of our proposed approach.

Maintaining neural plasticity represents a fundamental challenge in reinforcement learning, as agents must persistently adapt to dynamic environments. Recent studies(D'Oro et al., 2022; Schwarzer et al., 2023; Lee et al., 2024a; Ma et al., 2023; Nauman et al., 2024) have demonstrated that enhancing neural plasticity can lead to significant improvements in sample efficiency. Furthermore, both offline RL (Zhuang et al., 2023; 2024) and lifelong RL (Ahn et al., 2024) critically depend on maintained plasticity, though they present divergent challenges: static data exploitation in the former versus continual adaptation in the latter.

## 2.3. Policy Distillation in Reinforcement Learning

Policy distillation transfers knowledge from an RL agent to a smaller network for improved efficiency (Rusu et al., 2015). Prior research (Igl et al., 2020; Lyle et al., 2022) has highlighted its utility in addressing potential generalization loss in deep reinforcement learning agents due to nonstationarity and overfitting. This suggests that policy distillation serves two main functions: transferring knowledge and enhancing generalization, aligning well with the goals of plasticity recovery such as preserving knowledge and mitigating primacy bias. A critical aspect of effectively implementing policy distillation is quantifying the quality of knowledge transfer, which necessitates appropriate divergence measures.

A study by (Martins et al., 2021) discusses two types of Kullback-Leibler (KL) divergence measures: Forward KL (FKL) and Reverse KL (RKL). The Forward KL divergence, $D_{\text{KL}}^{\rightarrow}$, weights the state space according to the teacher's policy, prioritizing learning in states where the teacher's policy is more probable. Conversely, the Reverse KL divergence, $D_{\text{KL}}^{\leftarrow}$, weights according to the student's policy, promoting exploration and robustness but risking neglect of some teacher-favored behaviors. Their expressions are:

$$D_{\text{KL}}^{\rightarrow}(\pi_1 \parallel \pi_2) = \sum_{s \in \mathcal{S}} \pi_1(s) \log\left(\frac{\pi_1(s)}{\pi_2(s)}\right) \tag{2}$$

$$D_{\text{KL}}^{\leftarrow}(\pi_2 \parallel \pi_1) = \sum_{s \in \mathcal{S}} \pi_2(s) \log\left(\frac{\pi_2(s)}{\pi_1(s)}\right) \tag{3}$$

To tackle the performance degradation challenge in plasticity recovery, we devise a weighted integration scheme that leverages the complementary characteristics of forward and reverse KL divergence.

# 3. Neuron Regeneration

Maximizing neural network capacity utilization during training is challenged by diminishing plasticity, a consumable resource that depletes as networks learn. This depletion stems from suboptimal configurations such as inappropriate activation functions(Abbas et al., 2023), poor data quality (Lee et al., 2024a), and backpropagation limitations (Dohare et al., 2024), resulting in biased learning and suboptimal performance.

While neuron reset techniques offer promising solutions for plasticity restoration (D'Oro et al., 2022), they introduce a critical challenge: the risk of performance degradation without proper implementation (Nauman et al., 2024). Direct parameter reinitialization, while effective for restoring plasticity, can destabilize network performance.

To address this trade-off between plasticity restoration and performance preservation, we propose neuron regeneration as a dedicated mechanism, defined as:

---

**Definition 3.1: Neuron Regeneration**

Given a neural network with parameters $\theta = \{\theta_1, \theta_2, ..., \theta_n\}$, where each $\theta_i$ represents the parameters of the $i$-th neuron, let $S \subseteq \{1, 2, ..., n\}$ be any subset of neuron indices. The neuron regeneration operation is defined as:

$$\theta' = NR(\theta, S) = \{\theta'_1, \theta'_2, ..., \theta'_n\}$$

where:

$$\theta'_i = \begin{cases} \theta_i^{plastic}, & \text{if } i \in S \\ \theta_i^{new}, & \text{if } i \notin S \end{cases}$$

$\theta_i^{plastic}$ represents the reset plastic state and $\theta_i^{new}$ represents the potentially updated state of non-regenerated neurons. A neuron regeneration operation is considered effective if it maintains:
1. Plasticity Recovery: $\mathcal{P}(\theta') > \mathcal{P}(\theta)$
2. Performance Guarantee: $Per(\theta') \geq Per(\theta)$
where $\mathcal{P}$ denotes a measure of the network's plasticity and $Per$ represents the network's performance.

---

The proposed neuron regeneration mechanism should comprise two phases: a reset phase that initializes a subset of neurons $S$ to $\theta_i^{plastic}$ to enhance plasticity, followed by a recovery phase that adjusts the remaining parameters to $\theta_i^{new}$ to compensate for potential performance degradation. This dual-phase design aims to increase network plasticity $\mathcal{P}(\theta')$ while maintaining performance $Per(\theta')$. The adjustment of non-regenerated neurons preserves model stability, enabling sustainable plasticity for long-term learning at the neuron level and addressing traditional limitations in neural network training methods.

# 4. Methodology

To maximize the utilization of neural network capabilities, we propose the Sustainable Backup Propagation (SBP) method. SBP achieves sustainable plasticity through a novel neuron regeneration mechanism, which combines two key components: Cycle Reset and Inner Distillation.

## 4.1. Sustainable Backup Propagation

Our neuron regeneration mechanism integrates two complementary components: Reset and Distillation. Reset rejuvenates inactive neurons by resetting their parameters to initial states, effectively discarding outdated knowledge. Distillation preserves valuable information by transferring knowledge from the pre-reset to post-reset network. Together, these processes enable neurons to regain plasticity while maintaining essential learned features. By ensuring effective regeneration of individual neurons, we can maintain network-wide plasticity.

To implement network-wide neuron regeneration, we introduce the Cycle Reset mechanism, governed by two key parameters: Reset frequency $F$ and Reset rate $p$. Every $F$ training steps, $p\%$ of neurons in each layer undergo reset and regeneration following a cyclical order. Due to this sequential reset pattern, neurons with the longest survival time - those that have undergone more update steps and potentially experienced reduced plasticity - naturally become the targets of each reset cycle. This systematic process, illustrated in Figure 2, continuously refreshes neurons across all layers throughout training.

Inner Distillation completes the neuron regeneration process initiated by Reset. As shown in Figure 2, before resetting, the current policy $\pi_\theta$ is copied to a temporary policy $\pi_{\text{tem}}$. Selected neurons then undergo reset, clearing outdated information. Subsequently, through temporarily freezing reset neurons, Inner Distillation enables knowledge transfer from $\pi_{\text{tem}}$ to non-reset neurons in $\pi_\theta$, with the flexibility to handle any number of reset neurons (even a single one). This neuron-specific distillation process complements the Reset operation, jointly achieving neuron regeneration that yields both enhanced plasticity and maintained or improved performance compared to the original network.

The integration of Cycle Reset and Inner Distillation creates an effective neuron regeneration framework. While Cycle Reset maintains network-wide plasticity through systematic neuron regeneration, Inner Distillation ensures the preservation of learned capabilities during this process. Through this balanced regeneration process, SBP enables neural networks to maintain suitable plasticity throughout training and ultimately maximize their learning capability utilization. The complete mechanism is formalized as the SBP algorithm, detailed in Algorithm 1.

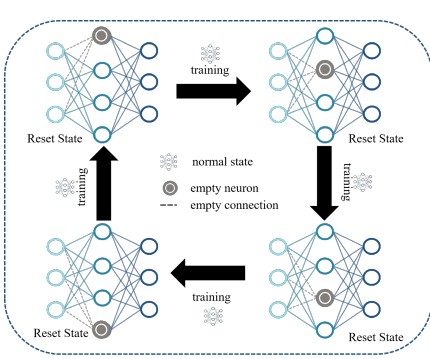 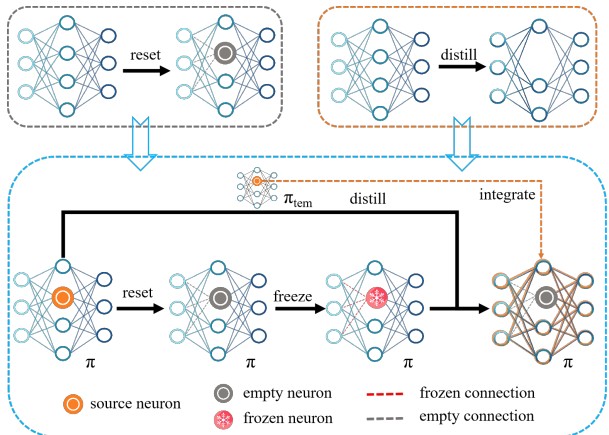

*Figure 2.* Left: Cycle Reset. Right: Inner Distillation. The right figure illustrates our method that combines neuron reset and knowledge distillation, where reset neurons restore plasticity while transferring knowledge to unreset neurons, thereby enhancing network plasticity while maintaining performance.

## 4.2. Double KL Divergence

Our analysis of Figure 1 revealed that the PPO algorithm faces challenges with plasticity loss. To address this issue, we propose integrating the SBP framework into PPO, aiming to enhance its capabilities. The Inner Distillation process presents a complex scenario where neurons may contain both valuable knowledge and irrelevant information. To maximize generalization and plasticity, we must carefully control this distillation process. In the context of reinforcement learning, we leverage the KL divergence as our distillation function, building upon previous research(Martins et al., 2021) that demonstrated the effectiveness of both Forward KL (FKL) and Reverse KL (RKL) in different aspects of knowledge transfer. Specifically, FKL has shown efficiency in transferring knowledge from a teacher policy to a student policy, while RKL is effective in preventing the infiltration of potentially harmful knowledge into the student model. To capitalize on the strengths of both approaches, we introduce a combined method. We also incorporate a parameter $\alpha$ to adapt the loss function to various distillation scenarios. This results in our proposed $\alpha$-weighted Double KL divergence ($\alpha$-DKL), expressed as:

$$\mathcal{L}(\theta) = \min_{\theta} \alpha \cdot D_{\mathrm{KL}}^{\rightarrow}(\pi_{\mathrm{tem}} \,\|\, \pi_{\theta'}) + (1-\alpha) \cdot D_{\mathrm{KL}}^{\leftarrow}(\pi_{\theta'} \,\|\, \pi_{\mathrm{tem}}) \tag{4}$$

- $\pi_\theta$ represents the primary policy, $\pi_{\mathrm{tem}}$ denotes a temporary policy typically used before a reset, and $\pi_{\theta'}$ signifies the policy after the reset of certain neurons.
- $D_{\mathrm{KL}}^{\rightarrow}(\pi_{\mathrm{tem}} \,\|\, \pi_{\theta'})$ denotes the Forward KL divergence, which measures how well the policy $\pi_{\theta'}$ approximates the policy $\pi_{\mathrm{tem}}$. This term is crucial for the effective transfer of essential knowledge from the $\pi_{\mathrm{tem}}$ to the $\pi_{\theta'}$.
- $D_{\mathrm{KL}}^{\leftarrow}(\pi_{\theta'} \,\|\, \pi_{\mathrm{tem}})$ represents the Reverse KL divergence, which acts as a regularizer to prevent the $\pi_{\theta'}$

from adopting potentially harmful or irrelevant information from the policy $\pi_{\mathrm{tem}}$.
- $\alpha \in [0, 1]$ is a tuning parameter that balances the contributions of the Forward and Reverse KL divergences to the overall loss function.

The $\alpha$-DKL approach offers a flexible and robust method for knowledge distillation, allowing us to balance the transfer of useful information with the prevention of harmful knowledge infiltration. By adjusting the $\alpha$ parameter, we can fine-tune the distillation process to suit different learning scenarios and optimize the trade-off between knowledge preservation and plasticity restoration. Leveraging $\alpha$-DKL, we propose Plastic PPO (P3O), an enhanced version of the PPO algorithm that integrates SBP and employs $\alpha$-DKL as its distillation loss function. This integration allows P3O to maintain sustainable plasticity throughout the learning process, potentially overcoming the limitations observed in standard PPO implementations. The details of P3O are presented in Algorithm 2.

## 5. Experiments

### 5.1. Experimental Setup

**Environment & Task** To evaluate our algorithm's performance, we employed a diverse set of tasks. These include standard benchmarks from MuJoCo (Todorov et al., 2012) and the state-based versions of DeepMind Control Suite (DMC) (Tassa et al., 2018). Additionally, we introduce the Cycle Friction Control task, an innovative variant of the MuJoCo environment inspired by the slip MuJoCo task (Dohare et al., 2024). Figure 8 shows a task with a cyclically changing friction coefficient. It starts at 4, decreases by 1 every million steps to 1, then increases back to 4. This discrete evolution significantly increases environmental complexity, challenging the algorithm.

*Table 1.* Performance comparison across MuJoCo environments (mean with 95% CI), with results averaged across 25 different random seeds. Percentages show improvement over baseline PPO. H-Stand: HumanoidStandup, Half: HalfCheetah.

|         | PPO | PPO+CBP | PPO+ReDo | PPO+Cycle | P3O |
|---------|-----|---------|----------|-----------|-----|
| Hopper   | 3719 [3668,3770] | 3692 [3653,3732] | 3730 [3672,3788] | 3532 [3258,3807] | **3765** [3699,3830] **(1%)** |
| H-Stand  | 135k [129k,140k] | 137k [133k,141k] | 141k [138k,145k] | 132k [122k,143k] | **144k** [136k,153k] **(7%)** |
| Walker   | 6270 [5974,6566] | 5284 [4991,5576] | 6240 [5891,6590] | 4993 [4761,5225] | **6788** [6380,6987] **(8%)** |
| Ant      | 3,354 [2839,3869] | 5480 [5377,5583] | 3579 [3318,3840] | 2210 [1820,2600] | **5522** [5421,5622] **(64%)** |
| Half     | 3626 [2450,4802] | 5944 [5687,6201] | 4936 [3663,6209] | 4585 [4025,5146] | **6865** [6181,7548] **(89%)** |
| Humanoid | 1163 [809,1518]  | 1966 [1196,2736] | 1413 [973,1854]  | 1066 [886,1247]  | **6237** [5901,6573] **(436%)** |

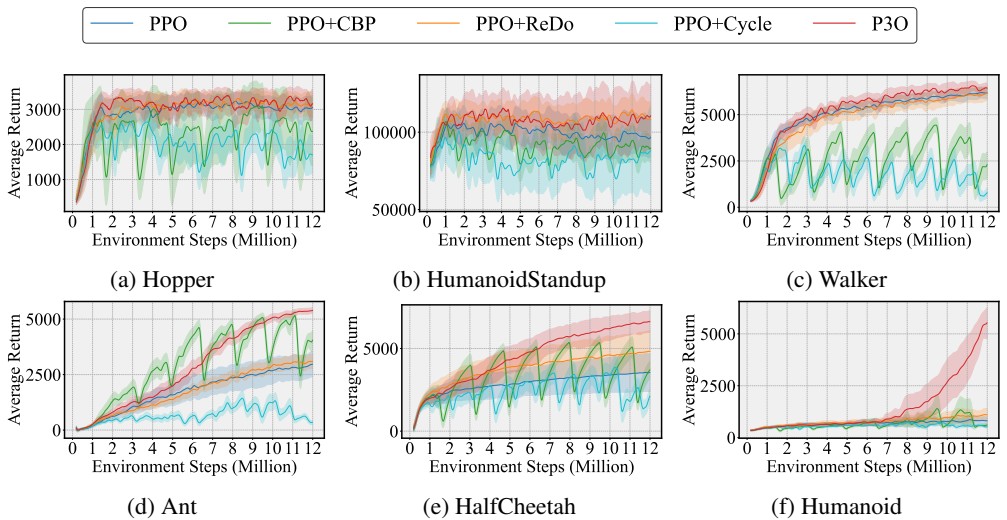

*Figure 3.* Performance of Various Algorithms in MuJoCo Environments

**Baseline** Throughout the entire experiment, we employed PPO as the base algorithm. In the reset experiment, we examined the impact of various reset strategies. **CBP** (Dohare et al., 2021) involves selecting neurons based on a utility function that considers both weight and activation values. **ReDo** (Sokar et al., 2023b) selects neurons based on a score derived from their activation values. **Cycle** involves selecting neurons in a specific order.

### 5.2. Experimental Results

**MuJoCo** Our experimental results demonstrate the effectiveness of our P3O algorithm across several key dimensions. Table 1 shows that P3O consistently outperforms other algorithms across six MuJoCo environments, with particularly remarkable improvements in the Humanoid environment, where the performance boost reaches 436%. This substantial increase in maximum rewards indicates that our framework effectively enables neuron regeneration, enhancing learning efficiency and more fully utilizing the neural network's capacity. The learning curves in Figure 3 further support this, demonstrating that P3O achieves the highest learning efficiency in most environments and maintains an upward trend for extended periods. This sustained improvement

suggests that we have indeed achieved sustainable plasticity, continuously providing the neural network with the capacity for learning and adaptation. Moreover, our ablation study using a standalone cycle reset without distillation demonstrates that knowledge recovery plays a crucial role in this process.

As illustrated in Figure 4, we analyzed the average L1 norm of network neuron weights during training. Our analysis reveals that while the original PPO algorithm tends to accumulate larger weights, algorithms incorporating resets maintain weights in a lower range. This observation aligns with recent research (Dohare et al., 2024) suggesting that excessively large weights indicate reduced neural plasticity. Our detailed analysis in the appendix (Table 4, Figure 16, and Figure 22) demonstrates a clear correlation between reset strategies and weight magnitude: both higher reset frequency and larger reset percentage contribute to smaller overall weights. P3O achieves a balanced weight distribution through its moderate reset frequency and full neuron reset strategy, positioning it between CBP's frequent partial resets and ReDo's limited reset approach. These findings confirm that strategic neuron resetting effectively restores plasticity by maintaining weights within an optimal range.

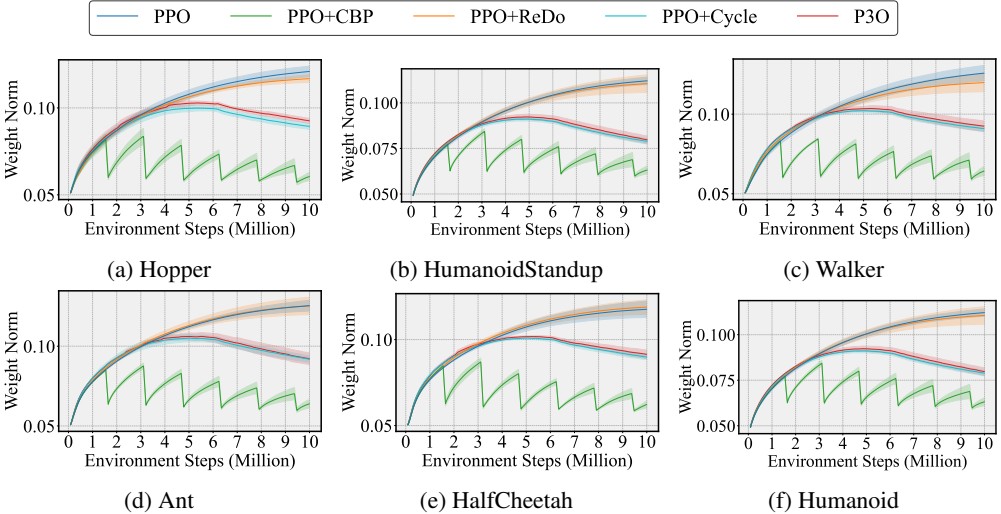

*Figure 4.* Actor Network Weight Norm (Lower norm tend to correlate with higher plasticity)

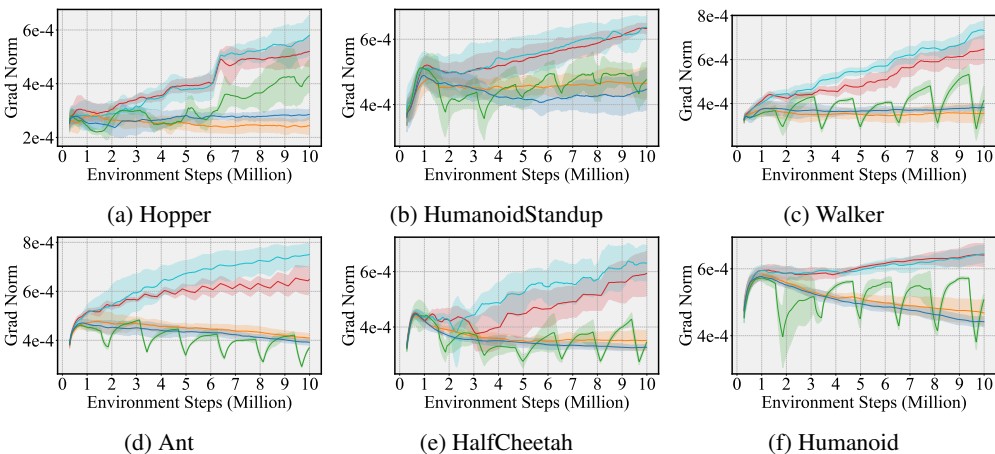

*Figure 5.* Actor Network Gradient Norm (Higher norm values tend to correlate with higher plasticity)

Beyond the benefits in weight regulation, Figure 5 reveals a crucial finding: P3O maintains consistently higher gradient norms than other tested algorithms. This is significant because learning occurs through gradients, and within a certain range, higher gradients indicate better learning efficiency and plasticity. While the phenomenon of vanishing gradients often signals a loss of plasticity, our algorithm effectively reduces weight values while maintaining robust gradients, resulting in optimal performance. Other algorithms show less effective gradient maintenance, suggesting the importance of knowledge retention in sustaining gradients. Further analysis in Figure 17 and Figure 23 reveals that both reset percentage and frequency significantly influence gradient maintenance.

The superior performance of P3O stems from its balanced approach: appropriate reset frequency combined with effective knowledge retention. This enables efficient neuron regeneration while effectively addressing the plasticity-

stability trade-off. By maintaining sustainable plasticity through systematic neuron regeneration, P3O achieves both smaller weight magnitudes and higher gradient norms, leading to enhanced sample efficiency and learning stability in reinforcement learning tasks. This demonstrates that neural renewal, when properly balanced with knowledge preservation, is key to achieving long-term adaptability.

**Advantages and Costs of Inner Distillation** Our distillation process utilizes an online replay buffer and operates in epochs. As shown in Table 5, the statistical results demonstrate the computational overhead of distillation. Our ablation studies about distillation are shown in Figure 3 and Figure 24. The statistics show that while inner distillation requires more training epochs, it achieves better performance. The results in Figure 3 and Figure 24 reveal that naive approaches to maintaining network plasticity are insufficient: cyclic reset without recovery operations leads to performance degradation, and simply increasing recov-

ery training epochs fails to maintain stable performance improvement. In contrast, our inner distillation approach effectively facilitates neural regeneration while preserving crucial knowledge, demonstrating both necessity and efficiency in knowledge retention.

As shown in Figure 25, our distillation-based approach achieves consistent performance improvements with increased epochs, while enhancing sample efficiency and effectively recovering historical knowledge. This stands in contrast to conventional approaches, where increasing training epochs often leads to performance degradation due to primacy bias (Nikishin et al., 2022), as observed in PPO implementations (Figure 1). Our approach, despite requiring additional distillation epochs, demonstrates superior performance over baseline approaches like CBP and ReDo, confirming its effectiveness in sustaining plasticity and ensuring stable learning.

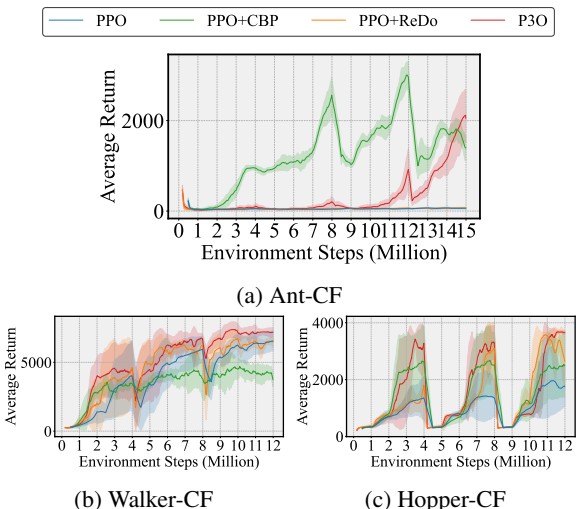

(a) Ant-CF

(b) Walker-CF          (c) Hopper-CF

*Figure 6.* Performance of Various Algorithms in Cycle Friction (CF) Environments

**DMC & Cycle Friction** Figures 7 and 6 reveal that in these complex benchmarks, while PPO struggles and other algorithms show only marginal improvements, our SBP approach achieves substantial progress. This superior performance in challenging environments demonstrates our algorithm's effectiveness and indicates that increased environmental complexity demands higher neural plasticity. Our method's success underscores the importance of effective neuron regeneration in complex tasks. These findings not only validate our approach but also highlight the need for further research into maximizing neural plasticity, especially in intricate learning environments.

Additionally, the experimental outcomes observed across Humanoid (Figure 3), Hopper Hop (Figure 7), and Cycle Friction Ant (Figure 6) environments demonstrate that the constrained plasticity of neural networks, rather than the

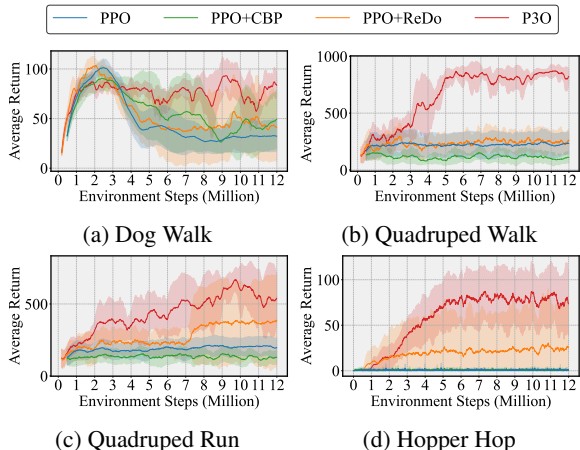

(a) Dog Walk          (b) Quadruped Walk

(c) Quadruped Run          (d) Hopper Hop

*Figure 7.* Performance of Various Algorithms in DMC Environments

inherent limitations of algorithms, often restricts the acquisition of valuable knowledge from data. This insight highlights the pivotal role of neural network plasticity in advancing learning capabilities.

## 6. Conclusion

In this work, we introduced the Sustainable Backup Propagation (SBP) framework, which addresses the fundamental challenge of maintaining neural network plasticity through systematic neuron regeneration. The core objective is to maximize the utilization of neural networks by providing sustainable plasticity while preserving their long-term learning capacity.

Our implementation, the Plastic PPO (P3O) algorithm, demonstrates the effectiveness of this approach across a diverse set tasks in deep reinforcement learning. Through its key components - a cycle reset mechanism that controls weight magnitudes and maintains gradient, combined with an inner distillation strategy that preserves valuable knowledge during neuron regeneration - P3O achieves both smaller weight magnitudes and higher gradient norms. These empirical results validate our hypothesis that proper neuron regeneration can create more adaptive AI systems capable of sustained learning over extended periods, effectively addressing the plasticity-stability trade-off in deep reinforcement learning by enabling networks to maintain high learning capacity while preserving acquired knowledge ("stay hungry, keep learning").

While our results demonstrate significant progress, several areas require further investigation, including the optimization of reset scheduling strategies and the theoretical analysis of neuron regeneration mechanisms. Nevertheless, our work provides new insights into maintaining network plasticity and establishes a foundation for future research in sustainable learning systems.

## Acknowledgments

This work was supported by the National Science and Technology Innovation 2030 - Major Project (Grant No. 2022ZD0208800), and NSFC General Program (Grant No. 62176215)

## Impact Statement

Our research, "Stay Hungry, Keep Learning: Sustainable Plasticity for Deep Reinforcement Learning," introduces novel approaches to address fundamental challenges in deep reinforcement learning, specifically focusing on sustainable plasticity and long-term learning capabilities.

The primary technical contribution, Sustainable Backup Propagation (SBP), represents a paradigm shift in how we approach neural network plasticity. By implementing a systematic neuron reset strategy combined with Inner Distillation, we demonstrate that networks can maintain learning capacity while mitigating performance degradation. This framework challenges conventional wisdom about conservative reset strategies and provides a practical solution for maintaining network plasticity throughout the learning process.

To validate our approach, we introduced the Cycle Friction Control task in the MuJoCo environment, which simulates real-world scenarios where environmental conditions change cyclically. This novel evaluation framework provides insights into the adaptability of reinforcement learning algorithms in dynamic environments, making our findings particularly relevant for real-world applications.

The broader implications of this work extend beyond technical achievements. By enabling more sustainable and adaptive learning systems, our research contributes to the development of more resilient and flexible AI systems. However, we acknowledge that enhanced plasticity in AI systems may raise important considerations regarding system stability and reliability. We encourage the research community to carefully consider these trade-offs when implementing similar approaches in practical applications.

We believe this work represents a significant step toward more capable and adaptable reinforcement learning systems, while recognizing the importance of balancing performance improvements with system stability and reliability. As the field continues to advance, we hope our findings will inspire further research into sustainable learning mechanisms and their practical applications.

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

# A. Appendix.

## A.1. HyperParameter

- Python 3.8
- Pytorch 2.0.1 (Paszke et al., 2019)
- Gym 0.23.1 (Brockman et al., 2016)
- MuJoCo 2.3.7 (Todorov et al., 2012)
- mujoco-py 2.1.2.14

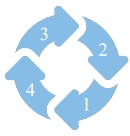

*Figure 8.* Cycle Friction.

Our experiment is based on PPO and incorporates SBP, CBP, and ReDo variants. We use the hyperparameters described in Table 2 for all algorithms. It's important to note that we've slightly modified the ReDo mechanism: instead of using a threshold-based selection, each reset is based on evaluating the scores of the bottom one percent of neurons. To ensure statistical significance and reproducibility, all experiments were conducted with 5 different random seeds, and the results presented are the mean values with corresponding standard deviations across these runs in figures. Figure 8 illustrates the pattern of friction changes of the cycle friction environments.

*Table 2.* Algorithm Parameters

| Category | Hyperparameter | Value |
|---|---|---|
| | Optimizer | Adam (Kingma & Ba, 2014) |
| | Learning Rate (Actor & Critic) | 3e-4 |
| | Online Replay Buffer Size | 8192 |
| | Mini-batch Size | 256 |
| PPO | Discount Factor | 0.99 |
| | Training Step | 1.5e7 |
| | Epochs per Update | 10 |
| | Clip Range | 0.2 |
| | Clip Grad Norm | 0.5 |
| | Actor & Critic Hidden Size | 256 |
| Architecture | Actor & Critic Hidden Layers | 3 |
| | Actor & Critic Activation Function | Tanh |
| | Reset Rate | 0.01 |
| | Reset Frequency | 50000 Environment Step |
| SBP | Neuron Utility Type | Neuron Lifetime |
| | DKL $\alpha$ | 0.4 |
| | Distillation Loss Bound $\tau$ | 0.01 |
| | Reset Rate | 0.01 |
| CBP | Reset Frequency | 10000 Gradient Step |
| | Neuron Utility Type | Contribution |
| | Reset Rate | 0.01 |
| ReDo | Reset Frequency | 50000 Environment Step |
| | Neuron Utility Type | ReDo Score |

## A.2. Concept

To systematically analyze plasticity, we introduce key terminology in Table 3. The emergence of primacy bias can be partially attributed to overfitting in neural networks. Dormant neuron identification requires careful hyperparameter selection, particularly:

- For ReLU activation: neurons producing zero outputs are considered dead

- For tanh/sigmoid: neurons with boundary-proximal outputs ($\approx \pm 1$ for tanh, $\approx 0$ or $1$ for sigmoid) are deemed saturated

The activation threshold for determining dormancy must be environment-specific, as it directly impacts plasticity measurements.

*Table 3.* Key Terminology Definitions

| Term | Definition |
| --- | --- |
| Plasticity | The ability of neural networks to learn from new experiences(Berariu et al., 2021). |
| Plasticity Loss | The diminished capacity of neurons to acquire new knowledge. (Lyle et al., 2023; Abbas et al., 2023) |
| Overfitting | The excessive fitting of a model to the training data(Zhang et al., 2018). |
| Primacy Bias | The tendency to overfit to earlier training data, resulting in poor learning outcomes on later sampled data(Nikishin et al., 2022). |
| Dormant | Neurons with low activation values in ReLU(Sokar et al., 2023a). |
| Dead (Saturated) | In ReLU activations, dead neurons occur when the output is zero for all inputs(Shin & Karniadakis, 2020; Lu et al., 2019). In sigmoid or tanh functions, neurons are considered saturated when the output approaches extreme values(Glorot & Bengio, 2010; Montavon et al., 2012). |

### A.3. Algorithm

In this section, we present two novel algorithms designed to achieve sustainable plasticity in neural networks: Sustainable Backup Propagation (SBP) and Plastic Proximal Policy Optimization (P3O). These algorithms address the challenge of maintaining neural network plasticity over extended periods.

#### A.3.1. SUSTAINABLE BACKUP PROPAGATION

Sustainable Backup Propagation (SBP) is a general framework designed to maintain the plasticity of neural networks over extended periods.By incorporating a neuron regeneration mechanism into standard backpropagation, it backs up sustainable plasticity for long-term network updates, hence enabling "sustainable" backpropagation.

The key components of SBP are:

- **Cycle Reset:** Periodically reinitializes a portion of neurons to prevent overspecialization.
- **Inner Distillation:** Utilizes a temporary model as a teacher network to recover essential knowledge after neuron reset.

This approach allows neural networks to maintain plasticity indefinitely, continually adapting to new information. The SBP algorithm implements the cycle reset and inner distillation mechanisms. In every F step, the reset rate $\gamma$ determines the proportion of neurons reset in each cycle, while the distillation process ensures that essential knowledge is retained after each reset. The distillation process continues until the distillation loss becomes less than $\tau$, indicating the selected neurons have successfully regenerated. The detailed algorithm is presented in Algorithm 1.

---

**Algorithm 1** Sustainable Backup Propagation (SBP)

---

Neural Network $f_\theta$, Temporary Model $f_{\text{tmp}}$, Reset Rate $\gamma$, Training Steps $T$, Reset Frequency $F$, Reset Index $p = 0$, Distillation Threshold $\tau$, Distillation Loss $d =$ None. **for** $t = 1$ *to* $T$ **do**

    Update Neural Network $f_\theta$ with standard backpropagation.

    **if** $t \mod F = 0$ **then**

        Copy the weights of Neural Network $f_\theta$ to Temporary Model $f_{\text{tmp}}$.

        **for** *each layer L of the Network* **do**

            Let $l =$ neurons of layer $L$.; Reinitialize input weights of neuron $i$ in layer $L$: $i \in [p \cdot l : (p + \gamma) \cdot l]$.

        Freeze all the reset neurons in Neural Network $f_\theta$.

        **while** $d > \tau$ *or* $d =$ *None* **do**

            Update Neural Network $f_\theta$ using Temporary Model $f_{\text{tmp}}$ as a teacher network, focusing on reducing the distillation loss $d$ according to a distillation loss.

            Recalculate Distillation Loss $d$.

        Unfreeze all the reset neurons in Neural Network $f_\theta$.

        **if** $p + \gamma < 1$ **then**

            $p = p + \gamma$;

        **else**

            $p = 0$;

---

#### A.3.2. PLASTIC PROXIMAL POLICY OPTIMIZATION

Plastic Proximal Policy Optimization (P3O) is a concrete implementation of the SBP framework within the context of reinforcement learning, specifically tailored for the Proximal Policy Optimization (PPO) algorithm. A specialized distillation function, DKL (Equation 4), is designed for PPO, ensuring effective knowledge transfer in policy space. The choice of PPO as our baseline implementation is particularly motivated by its inherent characteristics. As an on-policy algorithm maintaining only a small replay buffer, PPO faces significant plasticity demands due to the constant data distribution shifts inherent in reinforcement learning exploration. These characteristics make PPO one of the most challenging yet representative cases for testing plasticity mechanisms in reinforcement learning. We posit that if SBP can effectively enhance PPO's plasticity, it should generalize well to other reinforcement learning algorithms.

---

**Algorithm 2** Plastic PPO(P3O)

---

Policy $\pi_\theta$, Temporary Policy $\pi_{\text{tmp}}$, Reset Rate $\gamma$, Training Steps $T$, Reset Frequency $F$, Reset Index $p = 0$, Distillation Threshold $\tau$, Distillation Loss $d$ = None. **for** $t = 1$ *to* $T$ **do**

    Update Policy $\pi_\theta$ with regular policy gradient.

  **if** $t \mod F = 0$ **then**

      Copy the weights of Policy $\pi_\theta$ to Temporary Policy $\pi_{\text{tmp}}$.

      **for** *each layer L of the Network* **do**

         Let $l$ = neurons of layer $L$. Reinitialize input weights of neuron $i$ in layer $L$: $i \in [p \cdot l : (p + \gamma) \cdot l]$.

      Freeze all the reset neurons in Policy $\pi_\theta$.

      **while** $d > \tau$ *or* $d$ = *None* **do**

         Update Policy $\pi_\theta$ using Temporary Policy $\pi_{\text{tmp}}$ as a teacher network based on Equation 4.

         Update Distillation Loss $d$.

      Unfreeze all the reset neurons in Policy $\pi_\theta$.

      **if** $p + \gamma < 1$ **then**

         $p = p + \gamma$

      **else**

         $p = 0$

---

P3O demonstrates how the general SBP framework can be applied to specific machine learning paradigms, enabling sustainable plasticity in reinforcement learning policies. By incorporating neuron regeneration and knowledge distillation, these algorithms offer a promising approach to overcoming the limitations of traditional neural network training methods, particularly in scenarios requiring long-term learning and adaptation to changing environments. The detailed algorithm is presented in Algorithm 2.

### A.3.3. COMPARISON OF RESET STRATEGIES

In our study, we made deliberate choices in reset counting methods to align with each algorithm's characteristics while maintaining comparability. For CBP, we retained its original approach of using gradient steps for reset counting, preserving its algorithmic features. In contrast, for P3O, we opted to use environment interaction steps as the basis for reset counting. This decision was motivated by our focus on understanding how changes in input data affect neural plasticity. For consistency and to facilitate better comparison, we applied this same counting method based on environment interaction steps to ReDo as well. This approach allowed us to maintain the unique aspects of each algorithm while ensuring a meaningful comparative analysis across different reset strategies. Table 4 presents a comparative analysis of neuron reset statistics for the CBP, P3O, and ReDo algorithms throughout their respective training processes. The data represents the average across six Mujoco environments. Our findings reveal distinct patterns in reset frequency and scope among these algorithms:

- CBP exhibits the highest reset frequency, followed by P3O, with ReDo having the least frequent resets.
- In terms of reset scope, both CBP and P3O can reset all neurons, while ReDo has limitations in this aspect.

These reset patterns align with the weight norm distributions observed in Figure 4. The data suggests an inverse relationship between reset frequency and weight magnitude: more frequent resets correspond to smaller neuronal weights. This observation unveils a simple yet significant principle: the more frequent and comprehensive the resets, the smaller the neuron weights tend to be. However, excessively frequent resets can lead to performance instability. This trade-off suggests the importance of carefully calibrating both the reset frequency and recovery mechanisms to achieve an optimal balance between plasticity and stability.

*Table 4.* Reset Statistics Comparison (768 Neurons)

|       | Total Resets | Average Resets | Reset Proportion (%) |
|-------|--------------|----------------|----------------------|
| CBP   | 44208.6      | 57.6           | 100.0                |
| ReDo  | 1800.0       | 2.3            | 22.3                 |
| SBP   | 2304.0       | 3.0            | 100.0                |

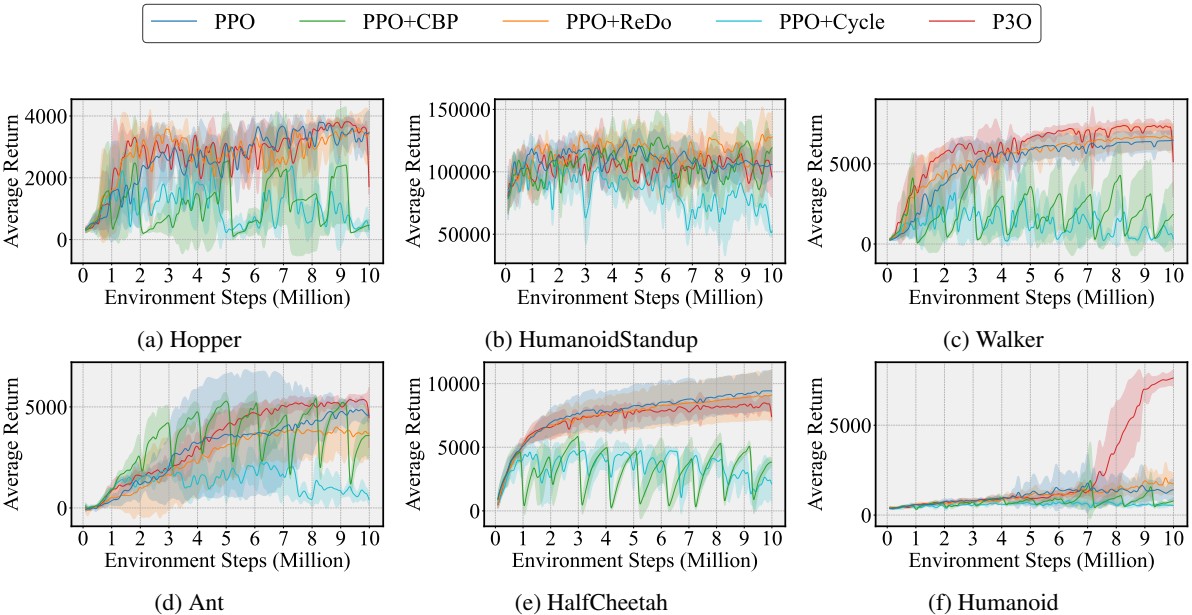

*Figure 9.* Performance Comparison of Different Algorithms with ReLU Activation in MuJoCo Environments

## A.4. Ablation

### A.4.1. ABLATION STUDY OF ACTIVATION

To comprehensively evaluate our algorithm's capability in maintaining plasticity, we conducted additional experiments across different activation functions, recognizing that activation is also a factor that affects the plasticity of neural networks. We maintained the same parameter settings as shown in Table 2 across all six environments. As shown in Figure 9 and Figure 3, our method demonstrates consistent performance improvements with both ReLU and Tanh activation functions, with the magnitude of improvement remaining comparable between these conditions. While baseline methods show strong performance with ReLU activation in some environments, such as HalfCheetah and Ant, they exhibit significant performance degradation when using Tanh activation. In contrast, our approach maintains stable performance across both activation functions. These results demonstrate that our algorithm exhibits superior robustness compared to existing methods, as it maintains consistent effectiveness regardless of the choice of activation function. This indicates that its plasticity-maintaining capabilities are not limited to specific activation functions but rather represent a more general and robust solution.

**Plasticity Exploration**

To better investigate the PPO algorithm, we chose the Tanh activation function based on previous research experience. Inspired by the neuron activation analysis in ReDo, we adopted a straightforward approach by using the absolute value of neuron activations as our scoring metric, leveraging the fact that Tanh activation values are bounded in [-1,1]. This score effectively reflects individual neuron activity levels, and we use the mean score across all neurons to quantify overall network activation. However, during our investigation of neural plasticity using dormancy rate as a metric, we made an unexpected discovery: networks using Tanh activation exhibited a clear correlation between activation function outputs and weight magnitudes. As shown in Figure 14 and 4, our analysis confirms a clear correlation between the overall magnitude of activation values and network weights under Tanh activation. This correlation suggests that ReDo's dormancy calculation method may not be directly applicable to networks using Tanh activation functions.

**Dormant Ratio** To address this concern, we conducted additional experiments with ReLU activation functions, with results shown in Figure 9, and calculated the dormant ratio using a threshold of 0.1 (Figure 10). The dormancy ratio curves closely align with performance variations - lower dormancy rates correlate with higher performance. Our method consistently maintains lower dormancy rates across most environments, following trends similar to those observed in weight and gradient norms (Figure 11 and Figure 12). However, the activation norm shows a more nuanced relationship. Comparing Figure 14 and Figure 13 reveals that both extremely high and low activation norms can be problematic. Our method consistently maintains stable activation norms across environments - approximately 0.7 for Tanh and 0.3 for ReLU activations. This

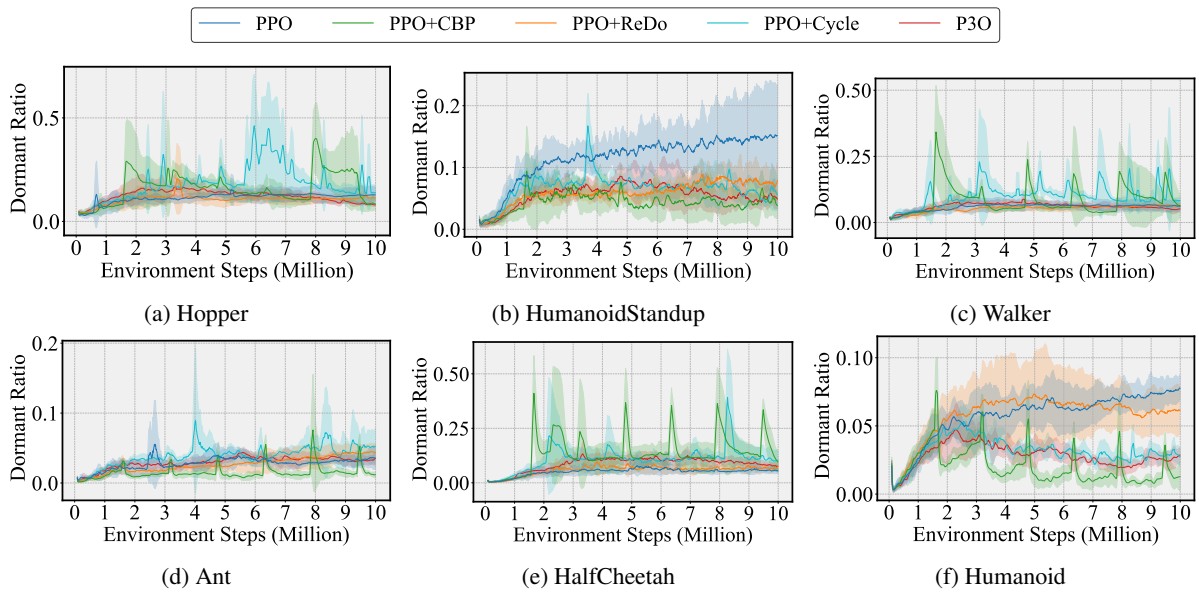

*Figure 10.* Dormancy Ratio of Actor Networks with ReLU Activation (Threshold = 0.1) across MuJoCo Environments

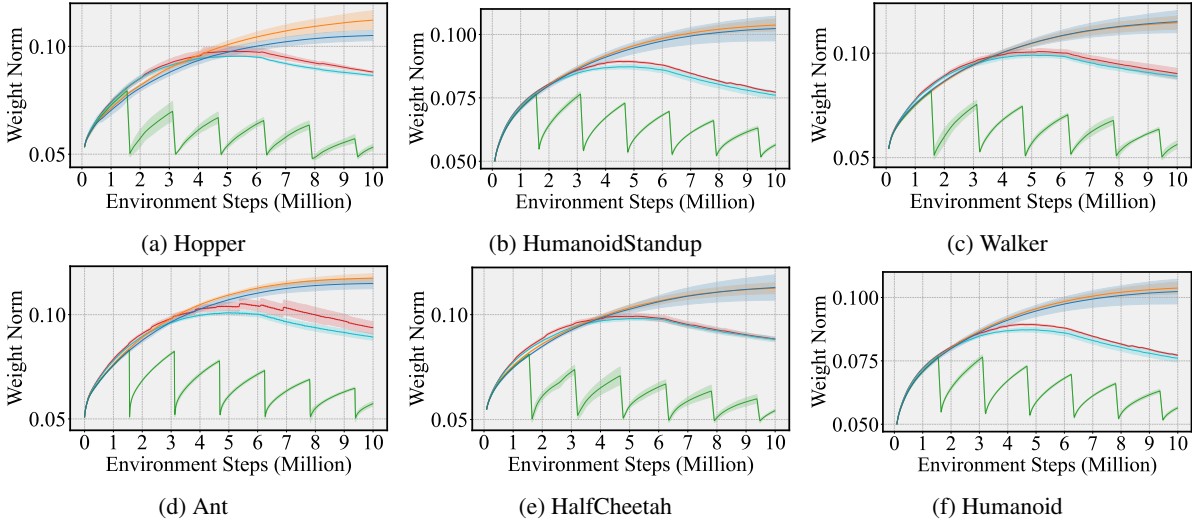

*Figure 11.* Actor Network Weight Norm with ReLU Activation (Lower norm tend to correlate with higher plasticity)

suggests that maintaining activation values within specific, activation function-dependent ranges might be crucial for plasticity, though this hypothesis requires further investigation. Therefore, weight and gradient norms serve as effective indicators of neural network plasticity and demonstrate the effectiveness of our algorithm. While dormancy ratio can also reflect plasticity levels, its applicability may be limited by the choice of activation functions. The relationship between plasticity and different activation functions warrants further investigation.

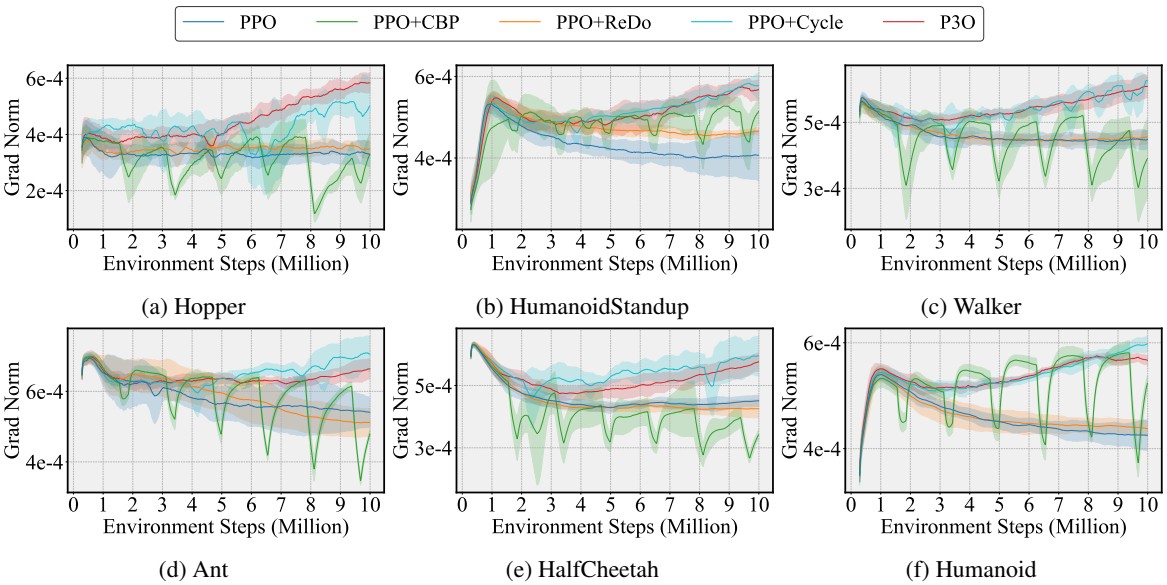

*Figure 12.* Actor Network Gradient Norm with ReLU Activation (Higher norm tend to correlate with higher plasticity)

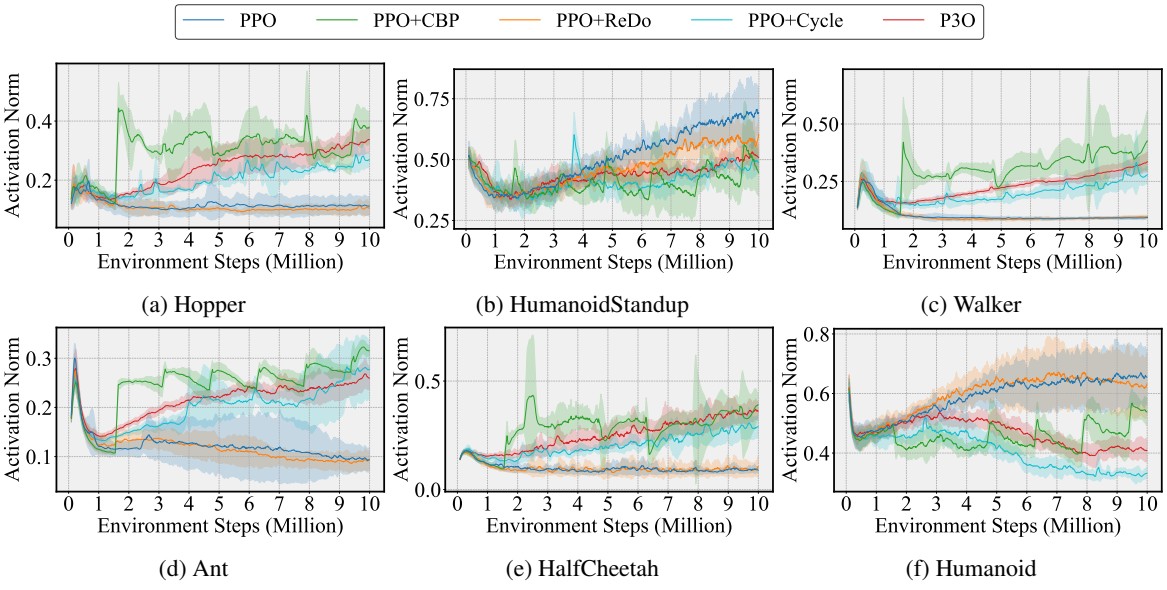

*Figure 13.* Actor Network Activation Norm with ReLU in MuJoCo Environments

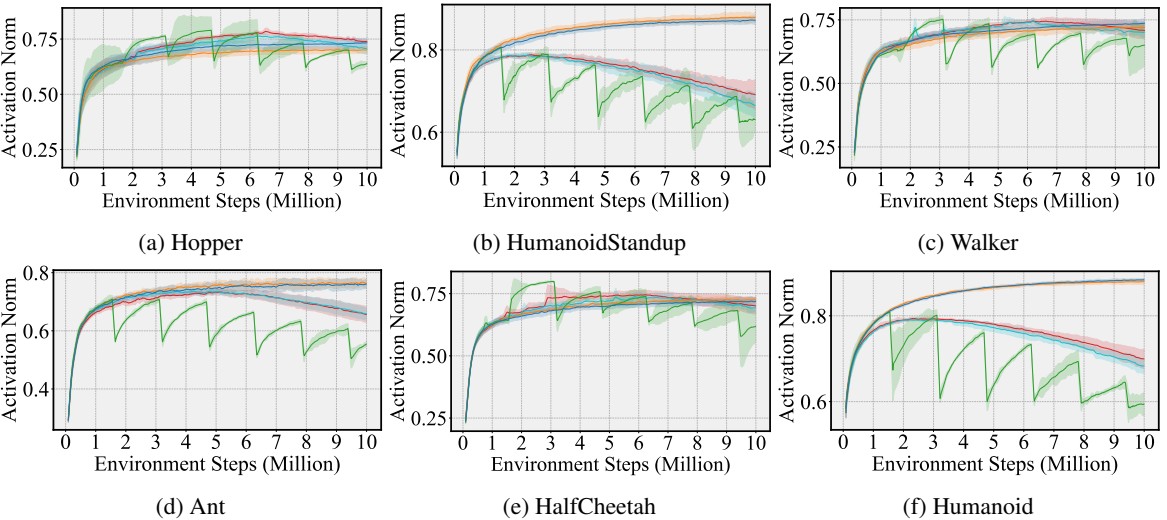

*Figure 14.* Actor Network Activation Norm with Tanh in MuJoCo Environments

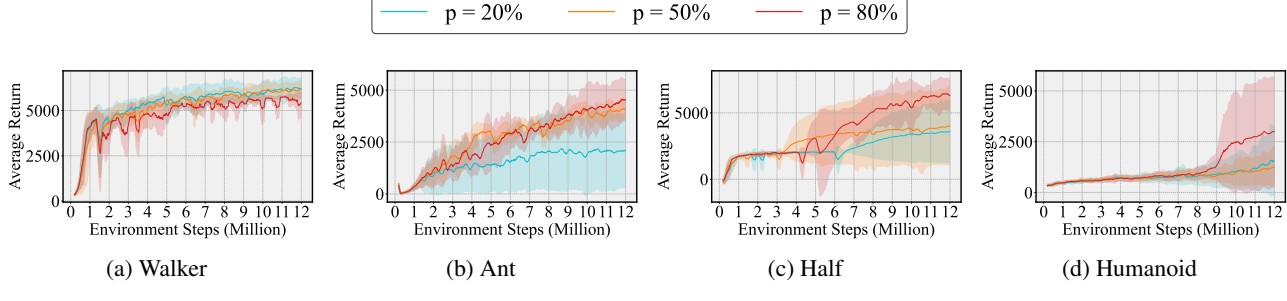

*Figure 15.* Performance of Various Network Reset Percentages

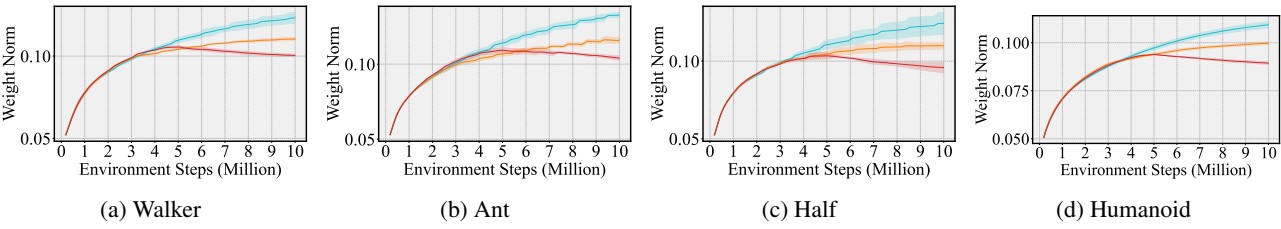

*Figure 16.* Actor Network Weight Norm Across Various Reset Percentages

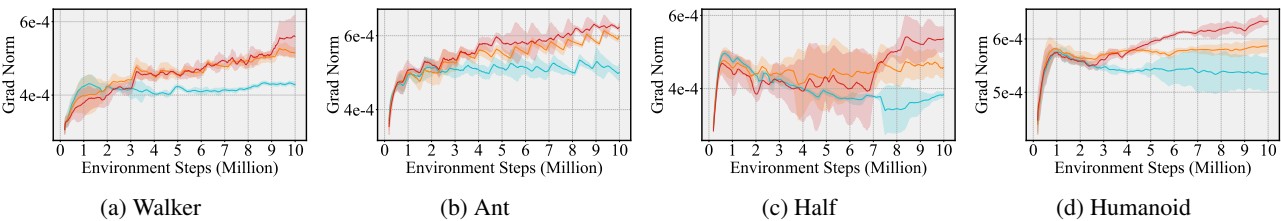

*Figure 17.* Actor Network Grad Norm Across Various Reset Percentages

### A.4.2. ABLATION OF CYCLE RESET

We present a systematic investigation of cycle reset's impact on neural plasticity through carefully designed ablation studies. Our experimental framework examines three fundamental parameters (reset percentage, rate, and frequency) and evaluates their effects through three complementary metrics: performance outcomes, weight norm distributions, and gradient norm patterns.

**Ablation of Reset Percentage**

In this section, we investigate how different reset percentages affect training performance to validate the necessity of cycle reset. Our experiments restrict neuron reset to the initial x% ($x \in 20, 50, 80$) of the network. Throughout training, we use a fixed per-reset rate of 0.01 with reset intervals of 50,000 steps, and cycle reset is applied only within the first x% of neurons. Figure 15, 16, and 17 show the experimental results for performance, weight variations, and gradient variations, respectively.

Our experiments reveal a clear correlation between reset percentage and network characteristics:

- In most environments, higher reset percentage lead to better performance
- Higher reset percentage result in smaller weight magnitudes across the network
- Networks with higher reset percentage maintain larger gradient norms throughout training

Our analysis of reset percentages reveals important insights into neural plasticity and neuron regeneration. Higher reset percentages consistently enhance network plasticity, as evidenced by weight magnitudes and gradient characteristics. Notably, while the Walker environment demonstrates increased plasticity with complete network reset, its performance trends differ from other environments. This divergence indicates that enhanced plasticity alone does not guarantee improved performance - successful regeneration requires appropriate recovery mechanisms to maintain learning stability. Our current

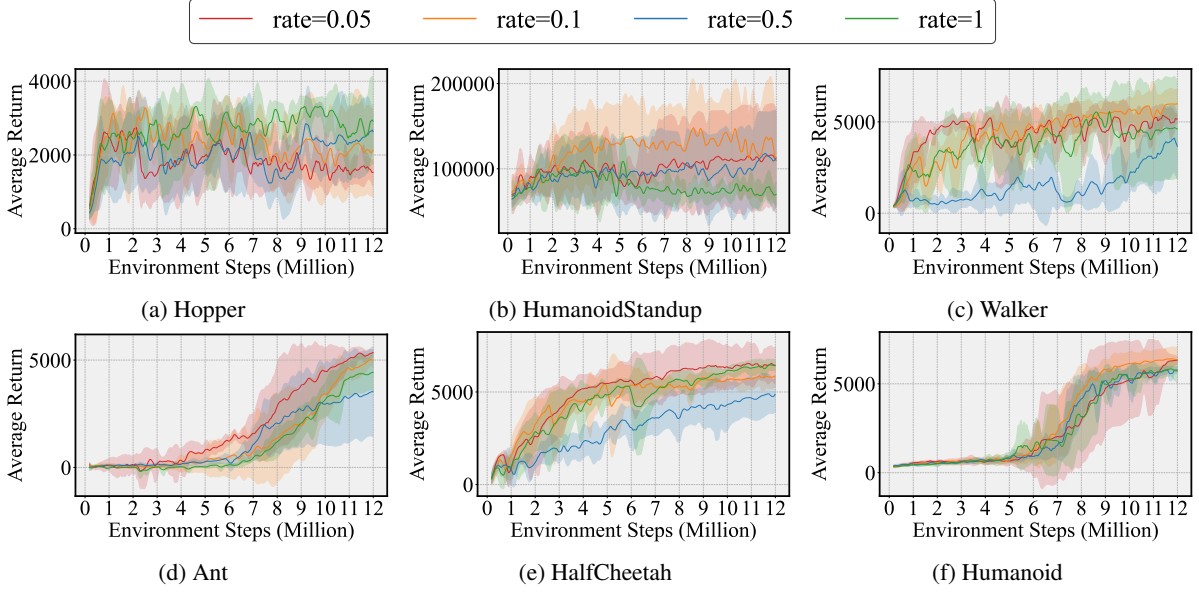

*Figure 18.* Performance Across Various Per-Reset Rates

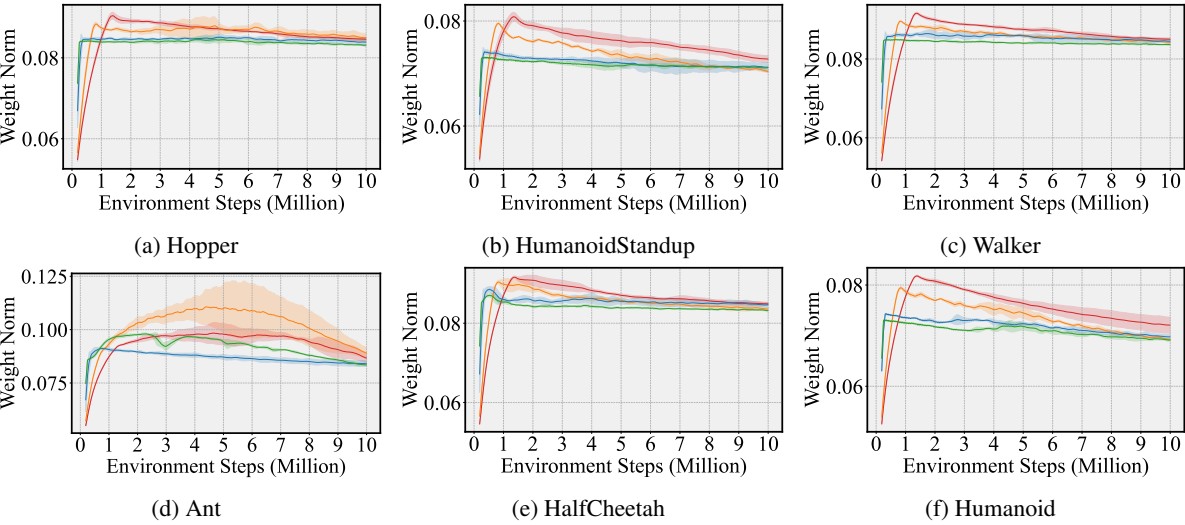

*Figure 19.* Actor Network Weight Norm Across Various Per-Reset Rates

parameter settings, while effective in other environments, require adjustment for the Walker scenario. These findings emphasize that effective neuron regeneration relies on carefully balancing network resets with recovery methods to achieve both optimal plasticity utilization and knowledge preservation.

**Ablation of Per-reset Rate**

To investigate the impact of reset rates on P3O, we conducted experiments with four different per-reset rate while maintaining a reset frequency of 50,000 environment steps. The results, illustrated in Figure 18 19 and 20, demonstrate several key findings:

- In most environments, reset rates of 0.05 and 0.1 achieved optimal performance.
- Resetting the entire network often outperformed resetting 50% of neurons.
- Weight magnitudes and gradient magnitudes reveals similar patterns across different per-reset rates.

This suggests that moderate reset rates can effectively achieve complete neural regeneration. Higher reset rates do not

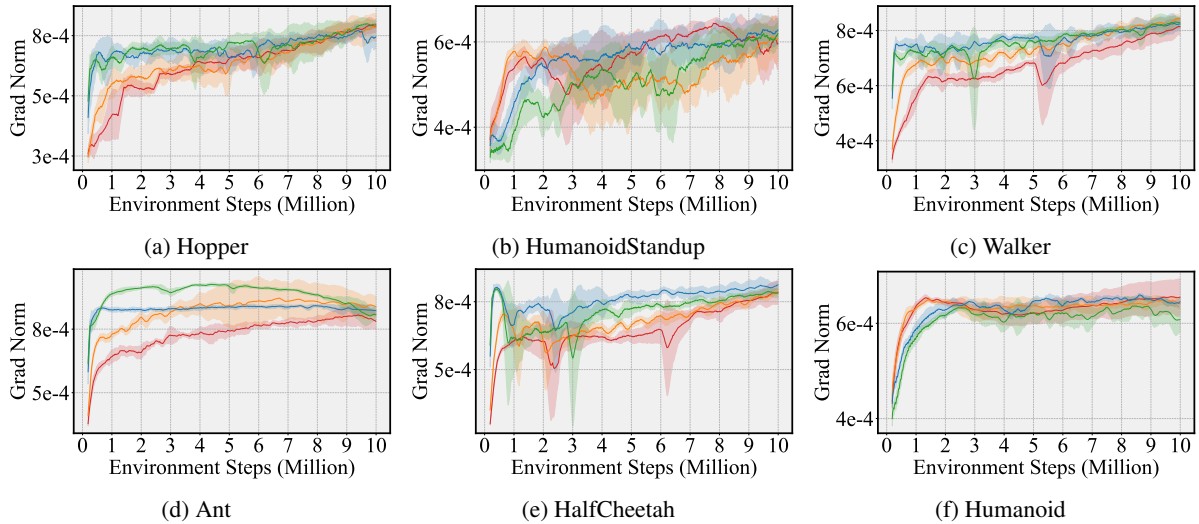

*Figure 20.* Actor Network Grad Norm Across Various Per-Reset Rates

necessarily lead to increased plasticity but instead create additional challenges for knowledge recovery, particularly within PPO's limited replay buffer setting. The superior performance of rate 1.0 compared to rate 0.5 is particularly evident in environments like Hopper, potentially indicating the influence of primacy bias on learning. These varying responses across environments suggest the need for more robust knowledge recovery mechanisms to handle different regeneration scenarios. Given that different reset rates yield similar plasticity improvements, implementing lower reset rates appears optimal as it minimizes knowledge recovery complexity while maintaining effectiveness.

**Ablation of Reset Frequency**

To investigate the impact of reset frequency, we conducted ablation studies while maintaining a fixed reset rate of 0.01. Figure 21, and presents our analysis across different reset intervals (20000, 40000, 80000, and 100000 steps). Our experiments reveal several key findings:

- Higher reset frequencies consistently lead to smaller weight magnitudes and larger gradient norms, indicating enhanced plasticity maintenance
- Higher reset frequencies generally correlate with better performance.

Reset frequency demonstrates a significant impact on neural plasticity restoration, with higher frequencies leading to enhanced plasticity maintenance. However, this relationship presents a trade-off: while more frequent resets maintain higher plasticity levels, excessively high reset frequencies can overwhelm the knowledge recovery process. This limitation potentially prevents the full conversion of increased plasticity into performance improvements. The results suggest the importance of finding an optimal reset frequency that balances plasticity enhancement with effective knowledge preservation.

Through our comprehensive ablation studies on cycle reset, we systematically investigated three key components: reset percentage reset rate and reset frequency. The experimental results reveal several crucial insights:

- **Reset Percentage:** Cycling through the entire network, rather than restricting resets to a subset of neurons, proves more effective in maintaining overall network plasticity.
- **Reset Rate:** Lower reset rates generally achieve optimal performance while maintaining stability.
- **Reset Frequency:** Higher reset frequencies correlate with enhanced plasticity, as evidenced by smaller weight magnitudes and larger gradient norms.

Based on these findings, we recommend a configuration with relatively small per-reset rates, higher reset frequencies, and complete network reset coverage. These results establish the effectiveness of our proposed cycle reset strategy while indicating directions for future research, particularly in developing more precise and adaptive reset schedules based on task characteristics and learning dynamics.

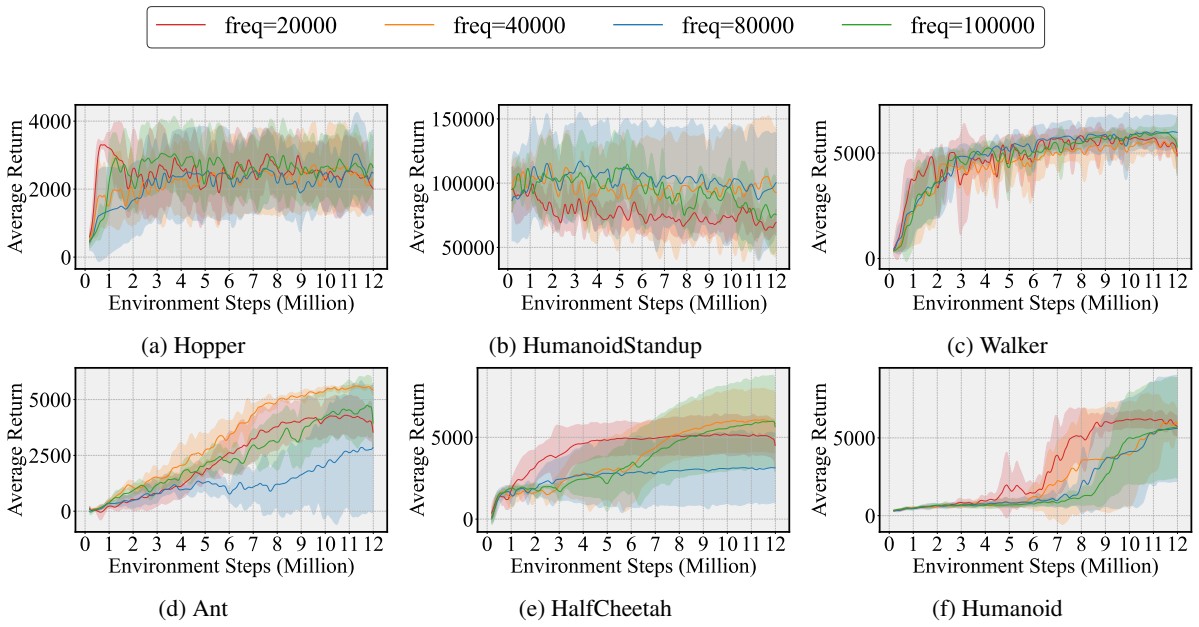

*Figure 21.* Performance Across Various Reset Frequencies

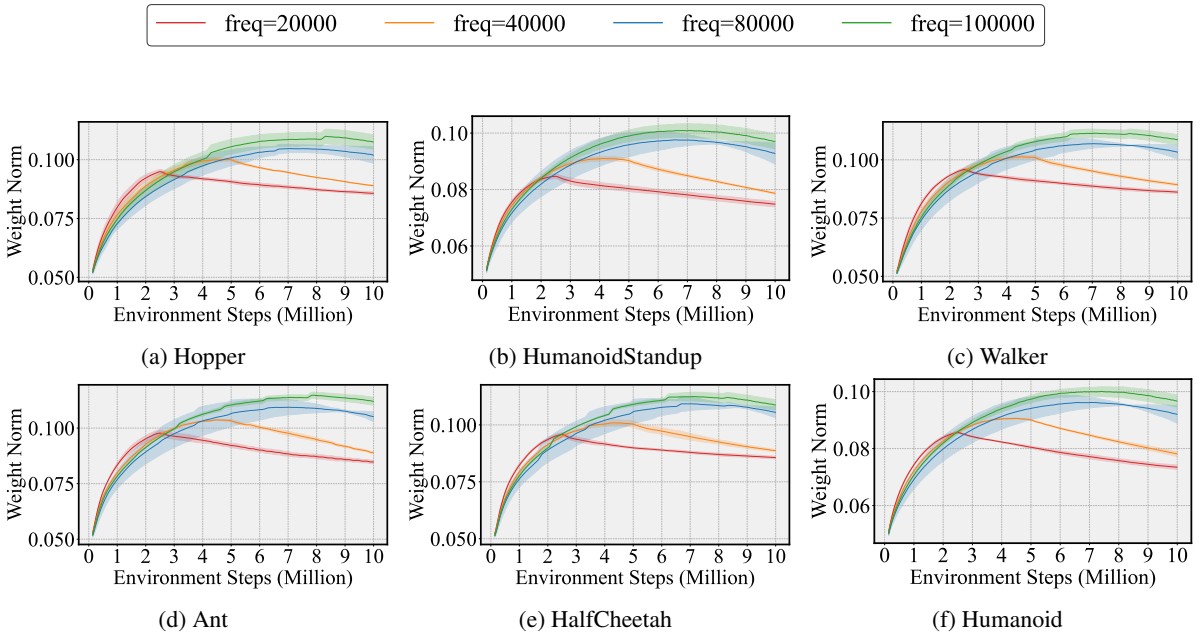

*Figure 22.* Actor Network Weight Norm Across Various Reset Frequencies

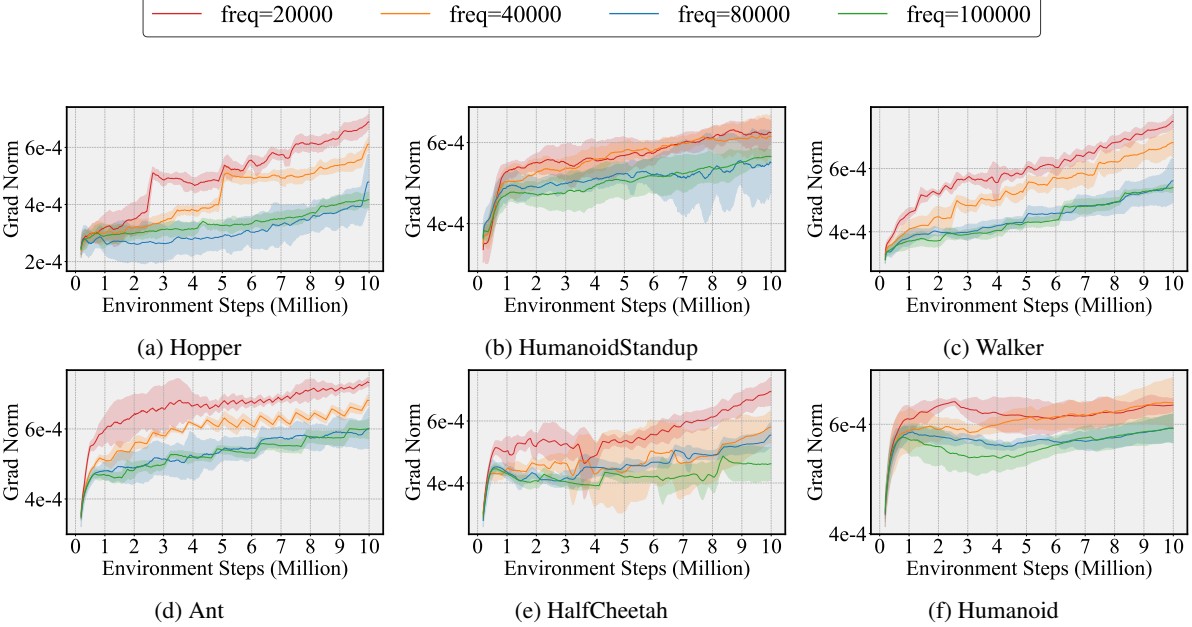

*Figure 23.* Actor Network Grad Norm Across Various Reset Frequencies

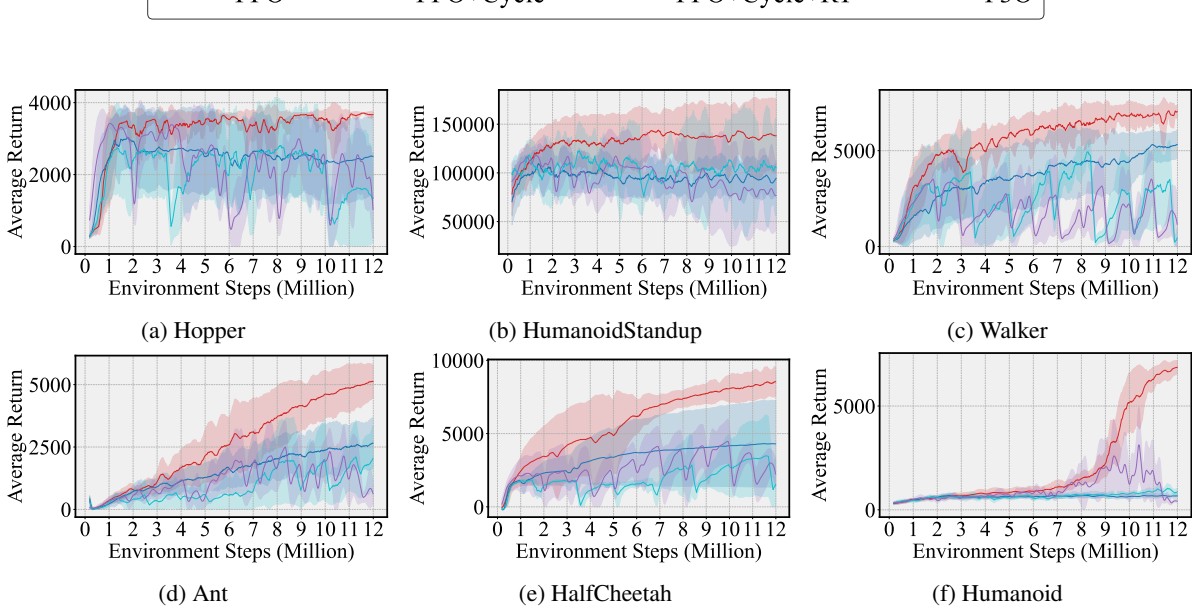

*Figure 24.* Performance Comparison with Recovery Training (RT)

### A.4.3. ABLATION OF INNER DISTILLATION

Through extensive ablation studies on cycle reset, we find that knowledge recovery settings play a crucial role in effective neuron regeneration. A robust knowledge preservation mechanism enables flexible neuron reset at arbitrary frequencies and proportions, facilitating effective regeneration of any selected neurons. Our investigation encompasses three key aspects: a comparative study between recovery training and knowledge distillation for knowledge preservation, an assessment of distillation robustness under different reset protocols, and a systematic exploration of the distillation hyperparameter $\alpha$ to optimize preservation performance.

**Inner Distillation VS Recovery Training**

To rigorously evaluate the recovery feature of our distillation mechanism, we conducted comparative experiments between inner distillation and simple extra training after cycle reset. We implemented a recovery training protocol where the network underwent additional training epochs post-reset until the reward difference between the recovered model and the pre-reset model fell below a threshold of 100, ensuring performance stability before proceeding to the next training cycle.

Our experimental results, as shown in Table 5 and Figure 24, demonstrate that while recovery training achieves moderate reward levels with lower computational cost compared to cycle reset alone, it fails to generate substantial performance improvements. This indicates that the ability to recover previous performance does not guarantee continued learning progress. In contrast, our distillation mechanism exhibits superior capabilities in both knowledge preservation and learning advancement. Notably, as illustrated in Figure 25, the additional epochs required for distillation translate directly into performance improvements, indicating meaningful training rather than computational overhead. These findings suggest that distillation not only effectively preserves and restores knowledge but also enhances sample efficiency by leveraging the network's plastic potential. The synergistic combination of reset and distillation thus emerges as a powerful approach for long-term learning improvement.

*Table 5.* Extra Training Epochs across Environments under 5 random seeds(PPO baseline: 18,310 epochs)

|  | Hopper | Humanoid Stand | Walker | Ant | HalfCheetah | Humanoid |
|---|---|---|---|---|---|---|
| Distillation Epochs | 597.66 | 638.19 | 1536.80 | 7698.40 | 2548.40 | 6525.25 |
| Recovery Epochs | 809.75 | 2731.60 | 524.00 | 700.20 | 253.20 | 485.25 |

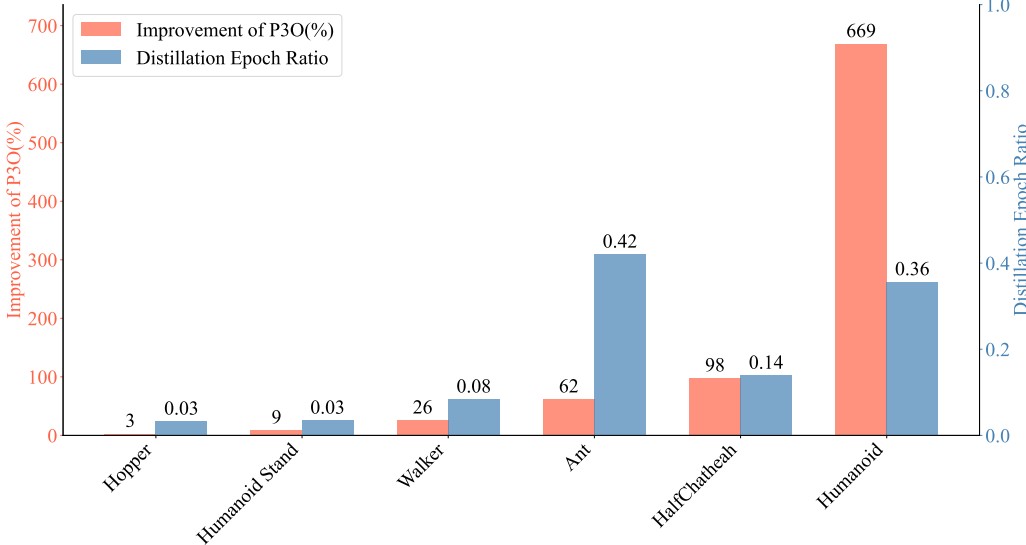

*Figure 25.* Cross-Environment Performance Gains vs. Distillation Epochs(under 5 random seeds)

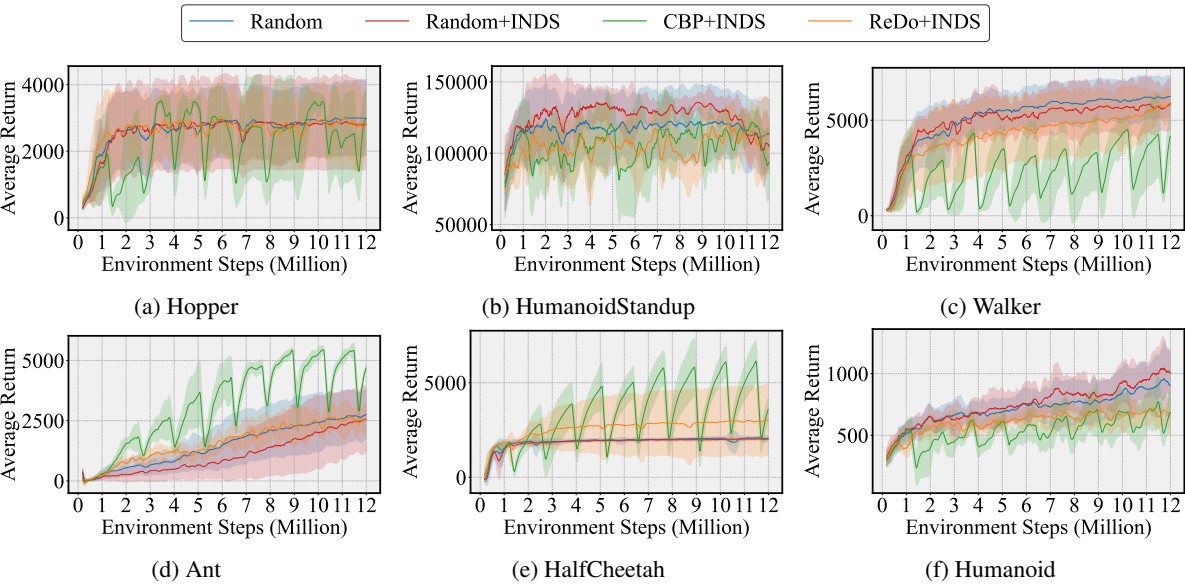

*Figure 26.* Performance of Inner Distillation(INDS) Integration with Different Algorithms across MuJoCo Environments

## Combining Inner Distillation with Baselines

Experiments combining inner distillation with various baselines were conducted to further investigate the efficacy of distillation-based knowledge preservation. As shown in Figure 26, distillation showed limited effectiveness when applied to two distinct reset strategies: Random, which randomly resets 0.01 of neurons, and ReDo. However, when integrated with CBP, distillation showed promising results, particularly in the Ant environment where notable performance gains were observed, despite some learning instability. It should be noted that these experiments utilized default parameters, which may have constrained the full potential of our algorithm's integration with CBP. These findings suggest two key implications: first, distillation can effectively complement existing reset strategies as a knowledge preservation mechanism in neuron regeneration paradigms; second, there remains significant potential for optimization through parameter tuning. Further research focusing on parameter optimization could enhance the robustness and effectiveness of combined distillation and neuronal reset approaches.

*Table 6.* Training Time Comparison (hours) across MuJoCo Environments

|  | Hopper | Humanoid Stand | Walker | Ant | HalfCheetah | Humanoid |
|---|---|---|---|---|---|---|
| PPO | 5.59 | 7.62 | 5.59 | 6.10 | 5.08 | 7.13 |
| P3O | 5.62 | 7.66 | 5.68 | 6.53 | 5.22 | 7.49 |
| Distillation | 0.03 | 0.04 | 0.09 | 0.43 | 0.14 | 0.36 |

In our experiments, we utilized a machine equipped with an NVIDIA V100 (32GB) GPU to measure the update time for the PPO, which averaged approximately 0.30 seconds per update epoch. For the distillation phases, we observed an average of 0.20 seconds per epoch, as these phases only require updating the actor network without the need to update the critic. This timing remains consistent across different environments. The differences in training times across environments primarily stem from variations in sampling times. However, since the distillation phases relied on PPO's own replay buffer, they did not require additional sampling. The training time of PPO consists of sampling time and update time, while P3O additionally requires distillation time. Ultimately, our results provide strong evidence that distillation does not significantly impact overall training efficiency, as demonstrated in Table 6. This suggests that the benefits gained from distillation in terms of performance do not come at a substantial cost to training time.

**Ablation of Distillation $\alpha$**

Parameter $\alpha$ modulates the knowledge transfer process during distillation, which critically influences the neural regeneration outcomes. Our experiments, as illustrated in Figure 27, suggest that different alpha values can affect learning efficiency. Our experiments demonstrate the effectiveness of Inner Distillation, with an optimal alpha value of 0.3 consistently yielding superior performance. This finding reveals that limiting forward-propagated information is beneficial and suggests the presence of primacy bias in current learning frameworks. The improved performance with a lower alpha value indicates that conventional learning paradigms may retain excessive redundant knowledge, highlighting the importance of effective neuron regeneration in balancing knowledge preservation and acquisition.

Our ablation studies validate Inner Distillation through three aspects: recovery training comparisons demonstrate its efficiency in performance restoration, tests across reset strategies confirm its generalizability, and analysis of parameter $\alpha$ reveals effective knowledge transfer control. These results establish Inner Distillation as an efficient and versatile approach for neural regeneration.

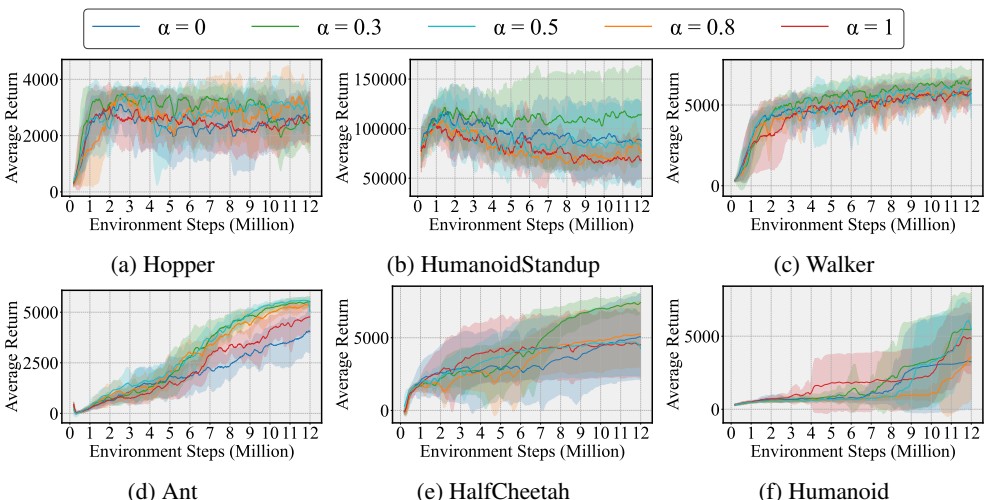

*Figure 27.* P3O Performance Sensitivity to $\alpha$ in MuJoCo Environments

### A.5. SAC with SBP

As a demonstration of our framework's generality in reinforcement learning, we extended our evaluation to Soft Actor-Critic (SAC)(Haarnoja et al., 2018), testing its performance across four Mujoco environments (Figure 28). We compared the ReDo, CBP, and Cycle reset approaches using the hyperparameter configurations detailed in Table 7. Additionally, we incorporated periodic resets of the last hidden layer and periodic resets of the entire network, which were studied in the context of off-policy methods.

Our results demonstrate that SBP consistently improves SAC's performance. The analysis of dormant ratio (Figure 29), gradient norm (Figure 32), weight norm (Figure 31), and activation norm (Figure 30) reveals trends similar to those observed in PPO: lower dormancy rates, larger gradients, smaller weights, and activation values maintained within stable ranges. Notably, despite a slightly higher dormancy rate in HalfCheetah, we observed larger activation norms, suggesting that similar performance was achieved with fewer active neurons, indicating more efficient utilization of neural plasticity. However, we believe these improvements represent only a fraction of SBP's potential benefit to SAC, particularly considering that our current implementation, which randomly samples just 1% (8,192 samples) from the replay buffer for inner distillation, already achieves significant plasticity enhancement and notable performance gains. The current approach, while effective, leaves substantial room for exploring more sophisticated sampling strategies to better utilize the rich information available in off-policy settings. To summarize, we believe that achieving better performance on SAC requires attention to the following two points:

**Distillation Buffer:** Constructing an appropriate distillation buffer is crucial. While the PPO online buffer can be directly used for distillation, SAC requires a suitable method to create a distillation buffer from its large offline buffer. An unsuitable buffer may exacerbate primacy bias.

**Impact of the Critic:** Our experimental results indicate that various reset strategies for the actor yield similar performance, suggesting that addressing the critic may be necessary to mitigate the loss of plasticity in SAC, an issue also discussed in previous work (Nikishin et al., 2022).

SAC faces challenges with a larger replay buffer, while PPO operates with a smaller one, leading to different exploration directions. Additionally, the critic seems to play a more significant role in the plasticity of SAC.

*Table 7.* Hyperparameter Configuration of SAC with SBP

| SAC Parameters | | SBP Parameters | |
| --- | --- | --- | --- |
| Activation Function | ReLU | Reset Frequency | 10,000 steps |
| Hidden Size | 256 | Reset Percentage | 0.01 |
| Training Step | 2e6 | Alpha Value | 0.8 |
| Replay Buffer Size | 1M | Distillation Buffer Size | 8,192 |

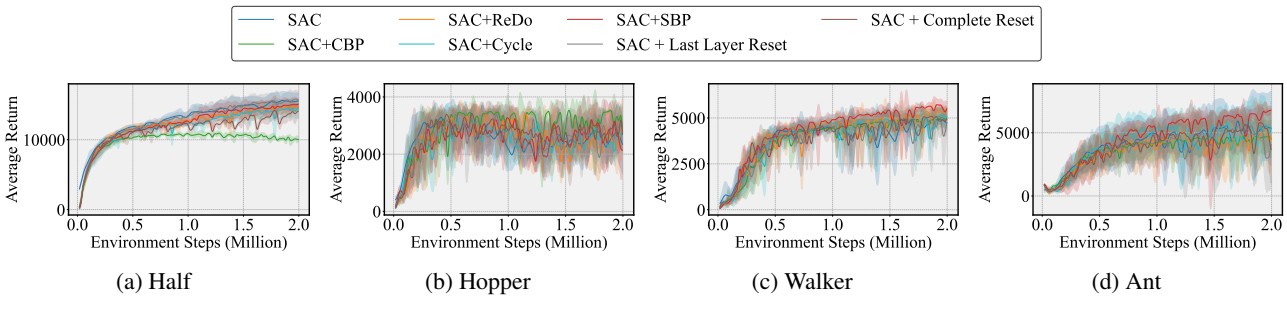

*Figure 28.* SAC Performance Across MuJoCo Environments

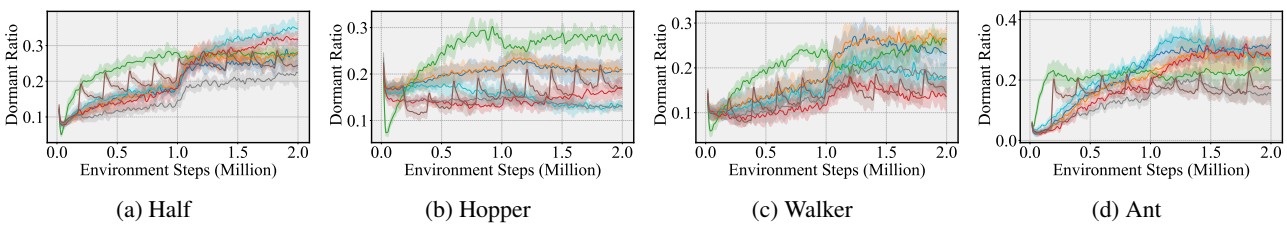

*Figure 29.* Dormant Ratio of SAC Actor Network in MuJoCo Environments (Threshold = 0.1)

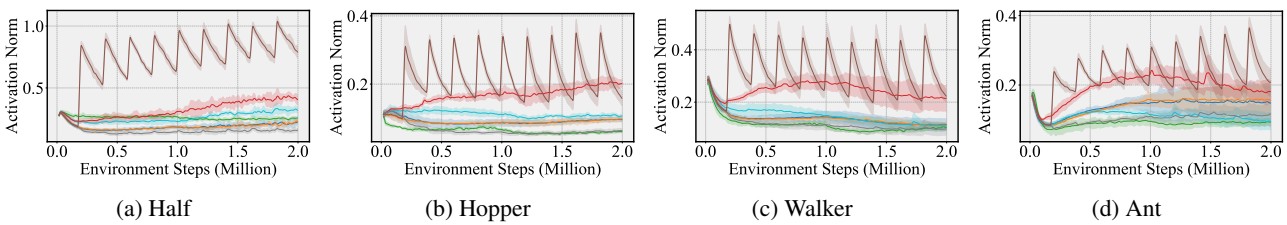

*Figure 30.* Activation Norm of SAC Actor Network in MuJoCo

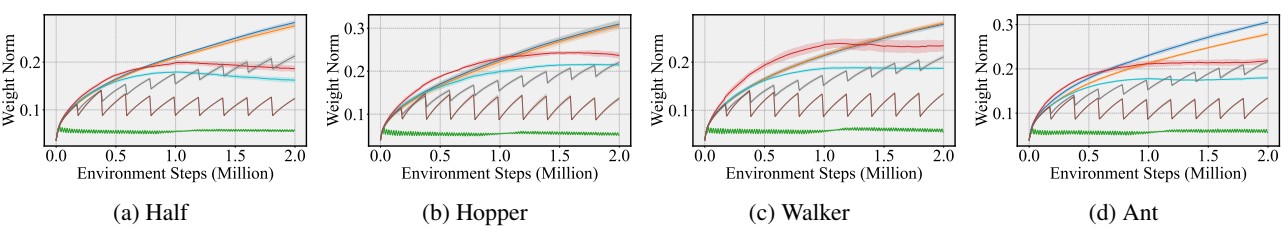

*Figure 31.* Weight Norm of SAC Actor Network in MuJoCo

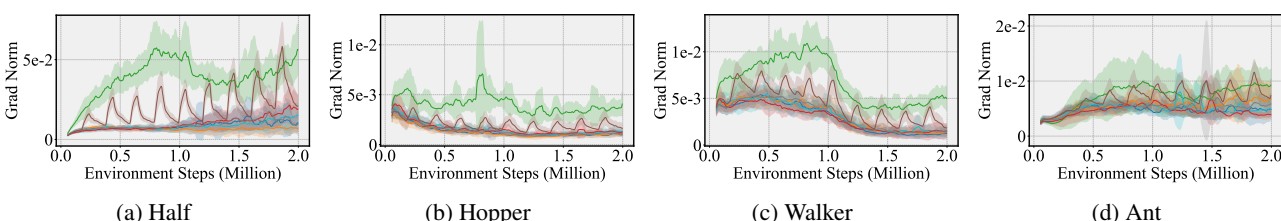

*Figure 32.* Gradient Norm of SAC Actor Network in MuJoCo

