# OpenReview forum: "Stay Hungry, Keep Learning: Sustainable Plasticity for Deep Reinforcement Learning"
_ICML.cc/2025/Conference — ICML 2025 poster_

### Official Review · Reviewer_Zet6 · 2025-03-13

**Overall Recommendation:** 1

**Summary:**

This paper proposes Plastic PPO (P3O) to address the plasticity loss in online RL. The key idea of P3O is the combination of cyclic neuron reset and inner distillation for policy network, which better balances the plasticity recovery and knowledge retention. The proposed methods are evaluated in MuJoCo and four DMC tasks, along with ablation study, hyperparameter analysis and other plasticity analysis in the appendix.

**Claims And Evidence:**

The scores in Table 1 do not have error bars. The results based on five seeds are not very convincing for PPO, as in Figure 3 the shaded areas overlaps a lot. Statistical significance is not mentioned.

**Essential References Not Discussed:**

An important related work that also proposed a reset-and-distill method is not included in this paper:

- Reset & Distill: A Recipe for Overcoming Negative Transfer in Continual Reinforcement Learning. arXiv 2403.05066

There are some other related papers on plasticity loss not included:

- Adam on Local Time: Addressing Nonstationarity in RL with Relative Adam Timesteps. arXiv 2412.17113
- Weight Clipping for Deep Continual and Reinforcement Learning. arXiv 2407.01704
- Directions of Curvature as an Explanation for Loss of Plasticity. arXiv 2312.00246

**Experimental Designs Or Analyses:**

The experiments for P3O include diverse aspects like performance comparison, hyperparam analysis, and ablation.

An issue is that the authors motivate this work with previous works on primacy bias, which are mainly studied for SAC with high UTD. However, in this paper, the proposed methods and the study of plasticity are for PPO, making it disconnected from previous literature on experimental study. For example, ReDO is originally proposed and evaluated for DQN and Atari. Reset is originally proposed for SAC. I noticed the experiments in Appendix A.4, but the proposed method does not show clear effectiveness upon SAC.

The motivating analysis in Figure 1 shows that PPO struggles leveraging more epochs due to plasticity loss, however, the direct empirical evidence for how P3O addresses this issue seems missing.

The direct comparison in terms of wall-clock training time or GPU hour time between baseline methods and the proposed method is missing, which is significant to practical use.

**Methods And Evaluation Criteria:**

The ideas of the proposed method make sense to me. The evaluation lacks sufficient random seeds and convincing statistics.

The motivating analysis in Figure 1 does not fully make sense to me. Since the failure of larger epoch numbers could also stem from increasing off-policyness of PPO training, the authors do not rule out this possibility in this paper.

**Other Comments Or Suggestions:**

- Equation 2 and 3 are a bit problematic in formal expression. I get the point here that $\pi_1, \pi_2$ are teacher and student respectively, but as they are placed in different orders, the formulas in Equation 2 and 3 are just normal KL.
- Line 23, “However, These”.
- In the first paragraph, spaces are missing between text and citations.
- Line 107, missing space for “P(” and a right “)” is missing too.
- Line 110, “learning, The goal”.
- Line 206, dual use of notation $P$, which denotes the transition probability above.

**Other Strengths And Weaknesses:**

- The name of the proposed method, Sustainable Backup Propagation (SBP), from my perspective, has nothing to do with how the method works, i.e., Cycle Reset and Inner Distillation.  I suggest the authors pick another name that reflects the proposed method directly. I felt a bit confused with the current method name SBP after reading Section 4.1.
- The paper writing needs substantial polish. I found many redundant and repeated expressions in Section 4. The content in Section 3 and Section 4 can be re-organized to be more connected and balanced.

**Questions For Authors:**

1. The distillation loss threshold $\tau$ is mentioned in Algorithm 1 and 2, but I did not find the detailed discussion nor the hyperparameter analysis experiment about it. Did I miss them?
2. How were the reset rate and the reset frequency selected? How will different choices influence performance?
3. In Line 185, the authors mentioned “we introduce the Cycle Reset mechanism, governed by two key parameters: Reset frequency F and Reset rate p”, but I found three factors in the appendix A.3.2, Reset frequency F, and Reset percentage, per-reset rate. After a quick check, I did find “per-reset rate” in the main text and Algorithm 1 and 2.
4. Although Figure 28 and Table 4 show the computation overhead of the proposed method, the direct comparison in terms of wall-clock training time or GPU hour time between baseline methods and the proposed method is missing.
5. As Figure 1 shows that PPO struggles leveraging more epochs due to plasticity loss, do P3O address this issue with direct experimental evidence?

**Relation To Broader Scientific Literature:**

The proposed method is related to the topics beyond plasticity loss, including continual RL, RL under non-stationarity.

**Theoretical Claims:**

Not applicable.

---

### Official Review · Reviewer_amiC · 2025-03-14

**Overall Recommendation:** 3

**Summary:**

The authors introduce the concept of neuron regeneration and propose a framework named Sustainable Backup Propagation(SBP) that maintains plasticity in neural networks through this neuron regeneration process. The SBP framework achieves network neuron regeneration through two key procedures: cycle reset and inner distillation.   The authors integrate SBP with Proximal Policy Optimization (PPO) and propose a distillation function for inner distillation. Experiments demonstrate the approach maintains policy plasticity and improves sample efficiency in reinforcement learning tasks.

## update after rebuttal
I want to update the score after reading the authors' responses to other reviews. There are weaknesses in the experiments and the literature has not been fully discussed.

**Claims And Evidence:**

The claims are clear and sound.

**Essential References Not Discussed:**

NA

**Experimental Designs Or Analyses:**

Experiments are solid.

**Methods And Evaluation Criteria:**

The methods and evaluation are solid.

**Other Comments Or Suggestions:**

None

**Other Strengths And Weaknesses:**

The paper was well-written and had a good structure, as well as extensive experiments.

**Questions For Authors:**

None

**Relation To Broader Scientific Literature:**

The method could potentially improve reinforcement learning.

**Theoretical Claims:**

There are no theoretical results. The paper could be improved with theoretical analysis.

---

### Official Review · Reviewer_4PTe · 2025-03-16

**Overall Recommendation:** 2

**Summary:**

This paper introduces Sustainable Backup Propagation (SBP), a framework designed to maintain neural network plasticity while preserving learned knowledge. SBP employs neuron regeneration through cycle reset and inner distillation and is integrated into Proximal Policy Optimization (PPO), leading to the development of Plastic PPO (P3O). Experiments in MuJoCo and DeepMind Control Suite show that P3O improves policy plasticity and sample efficiency compared to baseline methods.

Key contributions include:
- Neuron regeneration for sustained plasticity.
- SBP framework combining cycle reset and inner distillation.
- P3O as an enhanced PPO with SBP integration.
- Empirical validation demonstrating improved learning efficiency.

**Claims And Evidence:**

1. Neuron regeneration enhances plasticity: Supported by experiments showing controlled weight norms and higher gradient norms.
2. P3O Outperforms PPO and Other Baselines: Results in MuJoCo and DeepMind Control Suite demonstrate clear gains.
3. α-DKL Aids Knowledge Retention: Ablation studies confirm its role in balancing learning stability and flexibility​
4. Computational Overhead is Considered but Lacks Clarity: The paper compares distillation vs. recovery epochs (Table 4) and shows distillation improves performance despite extra training (Figure 27). However, direct runtime comparisons with PPO are missing​

The claims are well-supported, but clearer computational cost analysis would improve transparency.

**Essential References Not Discussed:**

The paper does not cite "Slow and Steady Wins the Race: Maintaining Plasticity with Hare and Tortoise Networks", a recent work on maintaining plasticity in reinforcement learning. Including this reference would help position SBP within the latest research on plasticity preservation.

[1] Slow and Steady Wins the Race: Maintaining Plasticity with Hare and Tortoise Networks., ICML 2024

**Experimental Designs Or Analyses:**

The experimental design is generally sound and aligns with standard reinforcement learning research methodologies:

- Multiple Environments: The inclusion of both standard and custom environments (Cycle Friction) strengthens the validity of the claims
- Ablation Studies: The paper conducts extensive ablations on reset frequency, reset percentage, α-DKL tuning, and alternative recovery methods, providing a comprehensive understanding of SBP’s impact

However, the paper does not thoroughly discuss the computational cost of SBP compared to PPO and its impact on training time.

**Methods And Evaluation Criteria:**

The proposed methods and evaluation criteria are generally appropriate for the problem at hand:

- Evaluation Benchmarks: The paper evaluates P3O in multiple RL environments (MuJoCo, DeepMind Control Suite, Cycle Friction) covering a range of tasks, ensuring a robust assessment​
- Baseline Comparisons: The comparison with PPO, CBP, and ReDo is well-structured, but additional baselines such as HnT[1] or recent plasticity-focused algorithms could further strengthen the analysis​

[1] Slow and Steady Wins the Race: Maintaining Plasticity with Hare and Tortoise Networks., ICML 2024

**Other Comments Or Suggestions:**

n/a

**Other Strengths And Weaknesses:**

n/a

**Questions For Authors:**

Recent works on plasticity preservation often evaluate methods in off-policy settings. Have you tested SBP in an off-policy reinforcement learning framework, such as SAC or DQN? If not, do you anticipate any challenges in adapting it to off-policy algorithms? A comparison in off-policy settings would help assess the generalizability of SBP beyond on-policy methods like PPO.

**Relation To Broader Scientific Literature:**

The paper builds upon prior work in reinforcement learning plasticity, reset mechanisms, and policy distillation.

**Theoretical Claims:**

No theoretical claims were made.

---

> ### Author Rebuttal · Authors · 2025-04-01
>
> We appreciate the reviewers' insightful comments and suggestions. Below are our detailed responses to the questions raised.
>
> ## 1. Reply to Questions 1
> Details about the off-policy (SAC + SBP) experiments are included in Appendix A.4. The results demonstrate performance improvements, and the plasticity metrics indicate favorable outcomes, highlighting the generalizability of our algorithm.
>
> ## 2. Reply to Computation Cost
> Since SBP operates as an independent plug-in outside the standard PPO process, it adds only one extra training step and utilizes PPO's existing replay buffer, incurring no additional sampling time. Consequently, the computational cost of P3O over PPO is primarily due to the overhead of inner distillation. The distillation epochs shown in Table 4 of Appendix A3.3 represent this additional computational cost. Given that the replay buffer and batch size remain consistent, we believe these epochs can be effectively converted into GPU hours, which is why we use epochs as the basis for our computational cost analysis.
>
> We will include a discussion and analysis of this aspect in the revised manuscript to provide clearer insights into the computational implications of our method.
>
> ## 3. Reply to References
> We will incorporate HnT into the related work section for further discussion.

---

### Official Review · Reviewer_2nT8 · 2025-03-23

**Overall Recommendation:** 3

**Summary:**

The paper tackles the problem of plasticity in on-policy reinforcement learning. Combining techniques of weight reseting and distillation, the authors propose a technique that the authors call "Sustainable Backup Propagation" (SBP). In SBP, some percentage of neurons are reinitialized every $n$ step. To mitigate the negative effects of neuron reinitialization, SBP maintains a copy of the pre-reset policy network which is distilled into the post-reset policy network using a weighted KL objective. The authors performed experiments on a few tasks from the OpenAI gym and DeepMind control suite.

**Claims And Evidence:**

> We introduce the concept of neuron regeneration, a biomimetic approach inspired by cellular regeneration processes...

A similar concept was introduced in previous work ("recycling dormant neurons") [1].

[1] Sokar, Ghada, et al. "The dormant neuron phenomenon in deep reinforcement learning." International Conference on Machine Learning. PMLR, 2023.

> We propose SBP, a systematic framework that implements neuron regeneration through cyclic reset strategies and inner distillation mechanisms...

This method seems novel and interesting. Combining resets and model distillation appears like a natural thing to do.

> By effectively addressing dead neurons and primacy bias, SBP ensures sustainable plasticity throughout the network’s lifecycle.

I think this claim should be slightly toned down. Figure 10 shows that the proposed method does not reduce the amount of dormant neurons as compared to the baselines. In general, I find the performance improvements convincing, but I feel that the authors provide a very limited analysis of why this might be (weight and gradient norm were shown to mildly correlate with plasticity in prior works).

> PPO that integrates SBP and a novel α-weighted Double KL divergence (α-DKL) loss function.

Again, calling weighted Jeffreys divergence novel is a bit of a stretch. Furthermore, whereas the authors provide some rationale for why this design choice might be important, there are no experiments ablating the importance of $\alpha$.

> Primacy bias/plasticity loss

I do not think this is established that primacy bias is the same as plasticity loss. I think the authors should provide more evidence that the primacy bias is at play (which was shown for SAC-based algorithms) or just stick to plasticity which is a more established umbrella term for learning problems.

**Essential References Not Discussed:**

The authors should certainly cite [3].

[3] Juliani, Arthur, and Jordan Ash. "A Study of Plasticity Loss in On-Policy Deep Reinforcement Learning." Advances in Neural Information Processing Systems 37 (2024): 113884-113910.

**Experimental Designs Or Analyses:**

1. Unconvincing choice of benchmarks - authors evaluate their method on OpenAI gym and DMC, benchmarks where off-policy methods are known to perform much better than PPO. Why not consider GPU-based simulators where PPO is dominant?

2. Missing baselines - recent work has shown that PPO+L2init and PPO+layer normalization are strong baselines when it comes to maintaining plasticity in on-policy algorithms

3. Limited ablations - whereas authors show that their proposed approach performs well, it is hard to attribute this performance improvement to design choices. For example, how important is $\alpha$ during distillation? Figure 26 suggests that the more resets the better, what would happen if we reset every gradient step? How costly is the distillation?

4. Are we sure this is plasticity at play? - the authors report results of 10mln environment steps training. Given the reported hyperparameters, this results in 800k policy updates during training. In contrast, in previous works, a single reset is performed every 2.5 min gradient steps. If authors claim that plasticity issues can occur in less than 800k gradient steps there should be some experiments to show it.

**Methods And Evaluation Criteria:**

In my opinion, the evaluation of the paper could be slightly improved. Whereas I appreciate the experiments on OpenAI gym and DMC, I think the authors should consider running experiments on environments where policy-based approaches like PPO perform relatively well (e.g. Isaac gym / Procgen).

Furthermore, I find it slightly surprising that the authors do not evaluate any off-policy methods - after all, approaches like ReDo or full-parameter resets were originally proposed for DQN and SAC-based algorithms. I think performing off-policy experiments for the proposed method on DMC/Gym would greatly enhance the presentation of the paper, as well as show the generalizability of the proposed method.

**Other Comments Or Suggestions:**

sometimes there is no space before the citation

**Other Strengths And Weaknesses:**

Strengths:

1. Plasticity studies for on-policy are important and potentially impactful
2. The proposed method seems to perform better than the evaluated baselines

Weaknesses:

1. Limited experimental section (only on-policy on a relatively small amount of environment steps). Does the method transfer to off-policy? How does the method pair with massively parallel simulations where plasticity might be more of an issue? These questions are unexplored.
2. Slightly confusing narrative of the paper: is inspiration by biological processes truly that important in this context? Is the paper tackling primacy bias plasticity or both or are they the same?
3. (nitpick) Figure 1 should be more related to the paper: the poor performance when increasing the number of epochs is not necessarily related to plasticity (e.g. might be that Q-values stop reflecting the policy). If it is indeed plasticity on the figure, does the proposed method allow for using more epochs?

**Questions For Authors:**

1. would the proposed method allow for more efficient learning in massively parallel setup?
2. would the method work for off-policy approaches like SAC?

**Relation To Broader Scientific Literature:**

The paper is highly related to previous works on the problem of plasticity (i.e. the decreasing capacity to adapt to new data as the learning progresses). The authors motivate their method by problems with existing solutions to plasticity such as the information churn stemming from full-parameter resets and propose a new method that is supposed to partly address these issues. What I find a bit confusing is that whereas most of the work in plasticity was done for off-policy algorithms, the authors focus on on-policy setup. While there is nothing wrong with that, it makes me wonder how general the insights proposed in this paper are.

**Theoretical Claims:**

NA

---

> ### Author Rebuttal · Authors · 2025-04-01
>
> We appreciate the reviewers' feedback and insightful comments. Below are our detailed responses to the questions and concerns raised.
>
> ## 1. Reply to Questions 1
> We believe the proposed method enhances learning efficiency in massively parallel setups. SBP acts as a flexible plug-in that integrates an independent module into the standard training process. Upon reaching the reset cycle, it transitions to inner distillation and then returns to normal PPO training, without affecting PPO's deployment.
>
> ## 2. Reply to Questions 2
> Details about the off-policy (SAC + SBP) experiments are included in Appendix A.4. The results show performance improvements, and the plasticity metrics also yielded favorable outcomes, indicating the generalizability of our algorithm.
>
> ## 3. Reply to Weaknesses 1
> Massively parallel setups may indeed face challenges related to plasticity. However, our work focuses on effectively maintaining plasticity to enable the network to learn stably over extended periods. We aim to address issues such as primacy bias, dormant neurons, and dead neurons while ensuring the stability of the neural network. This will enhance the algorithm's performance and unlock its potential. There is still much to explore in this area, and we believe that investigations into massively parallel setups may be better suited for future work.
>
> ## 4. Reply to Weaknesses 2
> Previous work related to reset mechanisms (Redo and CBP) has demonstrated that resetting can effectively alleviate the issue of plasticity loss. Our approach builds upon this reset foundation by introducing a recovery mechanism that ensures network stability. This not only addresses plasticity loss but also enhances the efficient utilization of plasticity, leading to improved sample efficiency and performance gains.
>
> Primacy bias refers to the fitting of early data, which hinders the ability to learn new knowledge later on. This concept aligns closely with plasticity, and prior work has categorized primacy bias as a form of plasticity loss. From both a conceptual standpoint and a consensus within the research community, primacy bias is recognized as a type of plasticity loss.
>
> ## 5. Reply to Weaknesses 3
> This can be addressed by referring to the response to review Zet6, point 5, regarding Figure 1.
>
> ## 6. Reply to Benchmarks
> This can be addressed by referring to the response to review wzCH, point 3.
>
> ## 7. Reply to Baselines
> Since we primarily focus on reset-based methods, we emphasize parameters related to resets, such as reset rate and reset frequency, as well as effective recovery strategies. Discussions on regularization methods are included in the related work section, as we consider them distinct types of approaches. While adding these baselines could further clarify our method's position in the broader field, we believe our current selection of baselines is appropriate for the questions we aim to explore.
>
> ## 8. Reply to Ablations
> Regarding the importance of the α parameter during distillation and the associated costs, our analysis of the ablation experiments is detailed in Appendix A3.3. We found that the maximum computational cost could exceed standard PPO by approximately 40%.
>
> As for the idea of resetting every gradient step, our conclusion is that more frequent resets result in smaller weights and maintain higher gradient levels, but do not guarantee stable performance improvements. This poses significant challenges for inner distillation, as each step must have accurately optimal parameters to ensure network stability and consistent performance improvement. This remains an area requiring extensive exploration.
>
> ## 9. Reply to Plasticity
> We believe that plasticity loss begins as soon as training commences, with plasticity being gradually consumed throughout the training process. Primacy bias emerges early on, while issues like dormant and dead neurons increase over time, all of which impact the ability to learn new information. This underscores the concept of plasticity. Additionally, Figure 1 illustrates the presence of primacy bias, and we contend that similar loss of plasticity occurs in PPO.
>
> ## 10. Reply to Some Claims
> Recycling only considers restoring neurons to a plastic state, without addressing the stability of the neural network. We believe that regeneration can encompass both aspects.
>
> "SBP ensures sustainable plasticity throughout the network’s lifecycle," and we will revise this to "SBP can sustainably provide plasticity and largely maintain network stability throughout the network’s lifecycle."
>
> Regarding the α-weighted Double KL divergence (α-DKL), our ablation experiments in Appendix A3.3 demonstrate that this weighting has a significant impact on the effectiveness of distillation. Therefore, we propose this loss function to be valuable and meaningful.

---

> > ### Comment · Reviewer_2nT8 · 2025-04-04
> >
> > Thank you for the rebuttal.
> >
> > > Details about the off-policy (SAC + SBP)
> >
> > Thank you for pointing me there - my bad for not noticing these results earlier. This is a nice start, though comparison to more SAC-native techniques like full parameter resets would help the reader to contextualize these results better.
> >
> > > Parallel environments
> >
> > GPU-based simulation makes it practical to run PPO for a lot longer than the 12 million steps considered in this paper. For example, recent works such as [1] run PPO for 1e^10 steps. I still believe that presenting results with lengthy training would greatly enhance the value of the paper - the results for dmc/gym can be considered good for on-policy, but mediocre when compared to off-policy algorithms.
> >
> > >  Currently, there is limited research on the plasticity of PPO, and no standard benchmark exists; previous studies [Dohare2024] have also been conducted in the same environments
> >
> > I agree that OpenAI gym is one of the standard benchmarks to test RL algorithms on. However, I think that there are substantial differences between the experimental setup presented in Dohare2024 that are relevant to the problem at hand - the cited work not only trains for ~4 times longer but also uses 4 times smaller batch size leading to a lot more gradient steps taken, which impact plasticity.
> >
> > Let me reiterate that I like the presented method and I think that studying plasticity in on-policy RL is valuable for the community. However, I think it slightly underdelivers its potential. What stops me from increasing the score at this point are the following doubts:
> >
> > 1. A bit unclear writing - authors use a variety of terms like plasticity loss, primacy bias, and dormant neurons, without clearly defining the relationship between them. For example: "As network neurons become saturated, they “become full”, losing the capacity to incorporate new information effectively. This reduction in plasticity (...). Additionally, the problem of overfitting in deep learning, known as primacy bias (Nikishin et al., 2022), further causes this loss of plasticity.". Whereas I agree that overfitting / plasticity / primacy bias / neuron dormancy are related, in my opinion, the way it is presented in the manuscript is confusing and does not help an uninitiated reader to better understand the relationship between the terms. My recommendation would be to present a table that defines these terms exactly, links to relevant literature, and perhaps discusses how these terms are similar and different.
> >
> > 2. Experimental setup - whereas running PPO for 12M steps on OpenAI gym was standard a few years back, I think it slightly disregards a lot of recent improvements the community made. For example, PPO run in GPU-based Mujoco Playground DMC performs on par with SAC, whereas the DMC results presented in the manuscript are "ok assuming low parallelization". This makes it hard to judge how the presented method fares in non-toy problems. Running experiments on a single GPU-powered environment leading to more competitive results would definitely make me improve my score.
> >
> > I hope that the authors can improve on the above aspects. At the same time, if other reviewers want to champion this paper in its current state, I will not be blocking such initiatives.

---

> > > ### Author Response · Authors · 2025-04-09
> > >
> > > # Response to Comments
> > > We would like to express our gratitude for your constructive feedback, which has significantly helped us improve the quality of our paper.
> > > ### 1. Reply to SAC
> > >
> > > We have built upon our original work by adding three baselines: CBP, periodic reset of the last hidden layer, and periodic reset of the entire network. The specific experimental results can be found in the following link: [Experimental Results](https://anonymous.4open.science/r/ICML_fig-0364/sac_fig.pdf). These results further clarify the differences between our algorithm and others within the context of SAC, helping readers better contextualize these findings.
> > >
> > > ### 2. Reply to Writing
> > >
> > > We appreciate your feedback regarding the clarity of our terminology. We recognize that terms like plasticity loss, primacy bias, and dormant neurons are interconnected, and we aim to clarify their relationships for readers who may be less familiar with the concepts.
> > >
> > > To address this, we have created the following table that clearly defines each term and highlights their relationships:
> > >
> > > | Term                | Definition                                                                                                       |
> > > |---------------------|------------------------------------------------------------------------------------------------------------------|
> > > | Plasticity          | The ability of neural networks to learn from new experiences.                                                   |
> > > | Plasticity Loss     | The diminished capacity of neurons to acquire new knowledge.                                                    |
> > > | Overfitting         | The excessive fitting of a model to the training data.                                                          |
> > > | Primacy Bias        | The tendency to overfit to earlier training data, resulting in poor learning outcomes on later sampled data.    |
> > > | Dormant             | Neurons with low activation values in ReLU.                                                                     |
> > > | Dead (Saturated)    | In ReLU activations, dead neurons occur when the output is zero for all inputs. In sigmoid or tanh functions, neurons are considered saturated when the output approaches extreme values. |
> > >
> > > Overfitting is a contributing factor to the emergence of primacy bias. The identification of dormant neurons depends on hyperparameters, specifically the activation value threshold set in a given environment; if the activation value falls below a certain threshold, the neurons are considered dormant. In the case of ReLU, dead neurons are those that output zero, while in tanh and sigmoid functions, they are considered saturated when the output is near the boundaries. We will include this table and discussion in the appendix and add relevant references to support these concepts.
> > >
> > > ### 3. Reply to Experimental Setup
> > >
> > > We would like to reaffirm that our work focuses on combining the reset mechanism with the recovery mechanism to establish a neuron regeneration mechanism. This leads us to propose SBP, which can effectively assist current backpropagation algorithms in maximizing the plasticity of each neuron.
> > >
> > > We demonstrate the necessity and effectiveness of SBP through P3O. Our extensive exploration and experimentation with PPO concentrate on plasticity metrics, and we provide detailed analyses of weight gradients and activations, which we believe strongly support our claims regarding SBP.
> > >
> > > We appreciate the reviewers' recognition of our algorithm's potential. We understand the desire for further exploration within PPO; however, due to the limited research in this area, we have struggled to find suitable references to justify more complex testing benchmarks. This challenge was evident during our initial research, where we could only select toy examples for validation. For future researchers in the community, our toy examples are necessary and valuable, as they can help avoid the challenges we faced and enable exploration of more complex benchmarks directly.
> > >
> > > Both long horizon and parallelization aspects present distinct challenges that require further exploration. Addressing long horizons necessitates more adjustments to PPO's parameters and corresponding changes to SBP parameters, which involves considerable effort. Furthermore, the Mujoco Playground DMC benchmark was only released after our paper submission, preventing us from including it in our experiments. We believe that, at this stage, our experiments are sufficient.
> > >
> > > While we acknowledge that we cannot encompass everything, our work provides important reference points for future research on plasticity in these areas. Our contributions are substantial and aim to advance the community's understanding while encouraging further investigations.
> > >
> > > ---
> > >
> > > We sincerely hope our responses have addressed your concerns. If there are no further questions, we would greatly appreciate the opportunity to improve our score.

---

### Official Review · Reviewer_UGKa · 2025-03-23

**Overall Recommendation:** 3

**Summary:**

The paper presents a new way of increasing plasticity in neural networks used in reinforcement learning. The main idea is to reset some of the neurons, while using a distillation strategy that maintains the "knowledge" of the reset neurons by the rest of the network. The method is considered in the context of the Proximal Policy Opitimization (PPO) and evaluated on a few RL environments.

## update after rebuttal
I am updating my initial score

**Claims And Evidence:**

1. It is not clear to me how widely applicable the methods are, what is their impact on off-policy methods?
2. Is there any impact of the method on the runtime of the baseline algorithm?

## update after rebuttal
Both points were addressed by the authors. I still think that the off-policy aspect requires more analysis, but I am happy about the presented direction in the supplementary material and authors responses.

**Essential References Not Discussed:**

No suggestions.

**Experimental Designs Or Analyses:**

I find the experimental designs valid.

**Methods And Evaluation Criteria:**

In general yes, but I have some doubts:
* what is the computational cost of the presented method compared to vanilla PPO? can it be ignored in the evaluation?
* would the same conclusions be made if the horizon of PPO training was longer?

**Other Comments Or Suggestions:**

* Definitions in section 2.1 seem not to be correct, e.g.:
    * the domain of the reward function is X x S, what is X?
    * P : S x A -> P, shoudn't it go to P(S)?
* The choice of using P vs \mathcal(P) in Definition 3.1 is suboptimal in my view - better to use something more descriptive (e.g., use subscript hinting at the meaning).
* section 4.2, you use \pi_temp and \pi_tem
* formatting in Table 1 is inconsitent (sometimes commas are used to separate thousands, sometimes not)

**Other Strengths And Weaknesses:**

Strenghts:
* clear idea, well presented,
* research is well motivated paper, plasticity is a very important aspect in RL

Weaknesses:
* why only PPO is considered, while other methods such as SAC (which is frequently used together with resets) is not evaluated?
* is the presented method slowing the baseline algorithm (PPO)? this is not evaluated in the paper

**Questions For Authors:**

* What exactly is presented in Fig 1? I don't fully understand those plots.
* Did you try to use your technique together with SAC or other off-policy algorithms?

**Relation To Broader Scientific Literature:**

The paper studies methods of improving plasticity in reinforcement learning. Losing plasticity is one of the well known problems of reinforcement learning, I find the study well motivated.

**Theoretical Claims:**

The presented results are mostly experimental. I did not find any issues related to formalizing the ideas of the paper.

---

> ### Author Rebuttal · Authors · 2025-04-01
>
> We sincerely appreciate the reviewers' feedback and insights. Below are our detailed responses to the questions and comments raised.
>
> ## 1. Reply to Questions 1
> Figure 1 demonstrates that primacy bias also exists in PPO. More training epochs lead to higher data fitting, but fitting the early data can hinder growth in later stages. This phenomenon was previously observed in SAC, and we believe it similarly affects PPO, which is the purpose of this figure.
>
> ## 2. Reply to Questions 2
> Details about the off-policy (SAC + SBP) experiments are included in Appendix A.4. The results show performance improvements, and the plasticity metrics also yielded favorable outcomes, indicating the generalizability of our algorithm.
>
> ## 3. Reply to Impact of Runtime and Computational Cost
> Since SBP operates as an independent plug-in outside the standard PPO process, it adds only one extra training step and utilizes PPO's existing replay buffer, incurring no additional sampling time. Consequently, the computational cost of P3O over PPO is primarily due to the overhead of inner distillation. The distillation epochs shown in Table 4 of Appendix A3.3 represent this additional computational cost. Given that the replay buffer and batch size remain consistent, we believe these epochs can be effectively converted into GPU hours, which is why we use epochs as the basis for our computational cost analysis.
>
> We will include a discussion and analysis of this aspect in the revised manuscript to provide clearer insights into the computational implications of our method.
>
> ## 4. Reply to Horizon of PPO
> For Ant, as shown in Figure 6C, at 15M steps, P30 is still increasing. In preliminary exploratory experiments, we ran up to 50M steps and found that P30 could reach 4000. However, due to the time required for these experiments, we did not conduct more to include them in the paper.
>
> ## 5. Reply to Typo
> We have corrected the formatting and typographical errors and thoroughly reviewed the paper to ensure clarity and precision.

---

> > ### Comment · Reviewer_UGKa · 2025-04-04
> >
> > Thank you for your clarifications, most of my questions are resolved, but there are still issues that I think should be covered in a revised version of the paper:
> > * wall-clock time impact (as also pointed by other reviewers),
> > * off-policy application - thank you for pointing me to A.4, but similarly as reviewer Zet6 I think more needs to be done on that front, especially when comparing against previous work.

---

> > > ### Author Response · Authors · 2025-04-08
> > >
> > > # Table 1: Training Time Comparison ((PPO baseline: 1,831 sample batch, 18,310 epochs (eps)))
> > >
> > > |                     | Hopper       | Humanoid Stand | Walker       | Ant          | HalfCheetah  | Humanoid     |
> > > |---------------------|--------------|----------------|--------------|--------------|---------------|--------------|
> > > | PPO Sample  (h)      | 4.07        | 6.10           | 4.07         | 4.58         | 3.56          | 5.61         |
> > > | PPO Update  (h)  | 1.52        | 1.52           | 1.52         | 1.52         | 1.52          | 1.52         |
> > > | PPO  (h) | 5.59        | 7.62           | 5.59         | 6.10         | 5.08          | 7.13         |
> > > | PPO + CBP (h)        | 5.59        | 7.62           | 5.59         | 6.10         | 5.08          | 7.13         |
> > > | PPO + ReDo (h)       | 5.59        | 7.62           | 5.59         | 6.10         | 5.08          | 7.13         |
> > > | P3O  (h) | 5.62        | 7.66           | 5.68         | 6.53         | 5.22          | 7.49         |
> > > | Distillation (h) | 0.03 (597.66 eps) | 0.04 (638.19 eps)  | 0.09 (1536.80 eps) | 0.43 (7698.40 eps) | 0.14 (2548.40 eps) | 0.36 (6525.25 eps) |
> > >
> > >
> > > ### Table 2: Sample One Batch (8192 Samples) Times Cross Environment
> > >
> > > |                     | Hopper | Humanoid Stand | Walker | Ant | HalfCheetah | Humanoid |
> > > |---------------------|--------|----------------|--------|-----|--------------|----------|
> > > | Sample Time (s)     | 8      | 12             | 8      | 9   | 7            | 11       |
> > >
> > > # Response to Comments
> > >
> > > We would like to express our gratitude to the reviewers for their valuable feedback and insights. Below are our detailed responses to the comments.
> > >
> > > ## 1. Reply to Wall-Clock Time Impact
> > >
> > > In our experiments, we utilized a machine equipped with an NVIDIA V100 (32GB) GPU to measure the update time for the Proximal Policy Optimization (PPO) algorithm, which averaged approximately 0.30 seconds per update epoch. For the distillation phases, we observed an average of 0.20 seconds per epoch, as these phases only require updating the actor network without the need to update the critic. This timing remains consistent across different environments.
> > >
> > > The differences in training times across environments primarily stem from variations in sampling times, as shown in **Table 2**. However, since the distillation phases relied on PPO's own replay buffer, they did not require additional sampling.
> > >
> > > The training time for PPO is the sum of sample time and update time. The same applies to CBP and ReDo, as they only introduce simple reset operations. In contrast, P3O incorporates the additional time required for distillation.
> > >
> > > Ultimately, our results provide strong evidence that distillation does not significantly impact overall training efficiency, as demonstrated in **Table 1**. This suggests that the benefits gained from distillation in terms of performance do not come at a substantial cost to training time. We will include this information in the appendix.
> > >
> > > ## 2. Reply to Off-Policy Application
> > >
> > > We have built upon our original work by adding three baselines: CBP, periodic reset of the last hidden layer, and periodic reset of the entire network. The specific experimental results can be found in the following link: [Experimental Results](https://anonymous.4open.science/r/ICML_fig-0364/sac_fig.pdf). These results further clarify the differences between our algorithm and other algorithms in the context of SAC, establishing a closer connection between our work and previous research.
> > >
> > > ---
> > >
> > > Thank you once again for your constructive feedback. We believe these additions and clarifications enhance the quality of our work. If there are no further questions, we hope for an increase in the score for our research.

---

### Official Review · Reviewer_XcpR · 2025-03-24

**Overall Recommendation:** 3

**Summary:**

This paper proposes a new Sustainable Backup Propagation(SBP) framework to maintain plasticity in Deep RL. SBP combines knowledge distillation with cyclical resetting of neurons. Results show that when SBP is combined with PPO, it results in much better performance and stability than PPO.

**Claims And Evidence:**

The claims made in this paper are not supported by clear evidence. This is an empirical paper. However, the empirical analysis is not statistically rigorous, and the results are not statistically significant. The authors study PPO and conduct five runs for all their experiments. This is not sufficient; PPO is known to be extremely noisy (Henderson et al., 2019). Experiments with PPO should have at least 30 runs. It is also not mentioned anywhere what is the shaded region in the plots. Is it the standard error, bootstrapped confidence interval or something else? The paper should report the 95% bootstrapped confidence interval for RL experiments. I suggest the authors read the paper by Patterson et al. (2024) on conducting proper empirical analysis in RL.

I am willing to increase the score if the authors conduct 30 runs and show that the conclusions stay the same.

Henderson et al., Deep Reinforcement Learning that Matters, 2019.

Patterson et al., Empirical Design in Reinforcement Learning, 2024.

**Essential References Not Discussed:**

N/A

**Experimental Designs Or Analyses:**

I checked the experimental designs and analysis. See above for the detailed issues with the analysis.

**Methods And Evaluation Criteria:**

The environments used in this study are appropriate.
It is unclear if other baselines (ReDo and CBP) were tuned appropriately. The paper does not contain any information on how the specific hyper-parameters for the baselines were chosen.

**Other Comments Or Suggestions:**

Please report the 95% bootstrapped confidence interval in all tables. Currently, Table 1 does not contain any confidence interval.

**Other Strengths And Weaknesses:**

The writing in some places is too strong and incorrect. For example, line 24 states that "these approaches ... fail to effectively reset the entire network, resulting in underutilization ...". This is written as a matter of fact. However, no evidence is provided for this claim. Similarly, line 45 states, " ... approaches, such as CBP (Dohare et al., 2021) ... focused on selectively resetting non-contributing neurons. While this strategy reduced information loss, it only partially restored plasticity ..." This is incorrect. Dohare et al. (2021) showed that CBP maintains plasticity in all cases tested. It is possible that CBP fails to maintain plasticity in some cases, but that needs to be shown before it can be claimed that CBP only partially restores plasticity. The caption of Figure 4 states," ... Lower norm indicates higher plasticity". That is incorrect; lower norms have been found to correlate with higher plasticity, but lower norms do not necessarily mean higher plasticity. The caption of Figure 5 has the same issue.

**Questions For Authors:**

What is the wall clock time for SBP compared to ReDo and CBP? It seems like SBP is signficaintly more computationally expensive than ReDo and CBP due to distillation.

**Relation To Broader Scientific Literature:**

Selective reinitialization is a common strategy to deal with loss of plasticity. This paper proposes a new method, SBP, to maintain plasticity using selective reinitialization. The results claim that SBP can outperform existing selective reinitialization methods like CBP and ReDo.

**Theoretical Claims:**

The paper does not contain theoretical claims

---

### Official Review · Reviewer_wzCH · 2025-03-28

**Overall Recommendation:** 3

**Summary:**

This paper addresses the loss of plasticity in deep reinforcement learning (DRL) models, which is a phenomenon where neural networks become less adaptable over time as they learn different task distributions. The authors explain that phenomena like primacy bias (overweighting early experiences) and dead neurons (units that cease to activate) are mainly responsible. The authors proposed a neuron regeneration mechanism inspired by cellular regeneration. The algorithm is called Sustainable Backup Propagation (SBP) that involves a periodic resetting of subsets of neurons and a distillation-based mechanism to  transfer knowledge from pre-reset neurons and preserve learning. The additional mechanisms are added on top of PPO, which is renamed as Plastic PPO (P3O). To manage a trade-off between preserving knowledge and adapting to new data, an additional mechanism named α-weighted Double KL Divergence (α-DKL) is introduced.

Experimental evidence  include  MuJoCo, DeepMind Control Suite, and a new Cycle Friction task). Baselines are two methods, CBP and ReDo, specifically designed to reinitialize neurons and mitigate the loss of learning ability of the network.

**Claims And Evidence:**

The paper claims that degrading learning is common in RL settings. This is reported in the literature and verified in the experiments showed in Fig 1.

The paper claims that the proposed approach ameliorates the issue of degraded learning. The experimental evidence seems to support this claim.

The claim that this algorithm implements sustainable plasticity is weak. The experimental settings do not involve sufficient distribution shifts to establish the algorithm's robustness.

**Essential References Not Discussed:**

The literature and approaches to lifelong learning are not discussed. While I understand this is a choice rather than an omission, I believe the paper would be stronger if the relationship between the proposed approach and established methods in lifelong learning was discussed.

**Experimental Designs Or Analyses:**

Overall, the paper presents a reasonable set of choices in the design of the experiments and the analysis. However, the evaluation over-emphasizes reset-based methods. I appreciate that this is a new emerging area to address degradation of learning, but I find it limiting that no other more general lifelong learning strategies are considered.

**Methods And Evaluation Criteria:**

In general, the methods and evaluation criteria are sound. However, the evaluation is on single-task only. Despite discussing lifelong/plasticity challenges, no sequential task benchmarks are used (e.g., Meta-World, continual Atari). No experiments on task transfer, catastrophic forgetting, or generalization across task changes. Thus, I believe the method is evaluated in a narrow context (single environment + cyclical changes) that limits the strength of the claims on plasticity in broader continual or real-world settings.

**Other Comments Or Suggestions:**

- Abstract, typo: However, These approaches

- Inconsistent spacing before a reference in the text, please check as sometimes comes with a space, sometimes without a space.

- The sentence "Additionally, the experimental outcomes observed across
Humanoid (Figure 3), Hopper Hop (Figure 6), and Cycle
Friction Ant (Figure 7) environments demonstrate that"
does not seem to match the figures: and-CF is in figure 6 not 7.

**Other Strengths And Weaknesses:**

Strengths:
- The paper addresses a known problem in neural network training, learning degradation when cycling through different distributions. This is an important limitation in continual learning systems, particularly applied to RL.
- The proposed method offers an interesting integration between reset approaches and continual learning with the inner distillation mechanism.
- The results seem favourable when compared with existing reset methods

Weaknesses:
- One main weakness in my opinion is the setup is far from a challenging continual learning scenario with multiple tasks and distribution changes. The paper is addressing a limited and constrained problem in which one parameter (friction coefficient) is responsible for the nonstationarity of the environment that can take 4 different states. Given the limited source of nonstationarity, I suspect that a  context-based meta learner, e.g. CAVIA or PERL, would perform well. I appreciate that those require task boundaries, but SBP goes around it with a periodic reset frequency, which is one additional hyperparameter.
In short:
- The approach is evaluated only in single-task, mostly stationary settings, with nonstationarity limited to a single engineered environment (Cycle Friction).
- Despite broader claims on plasticity and sustainable learning, there is no multi-task, continual learning, or transfer learning setup.

**Questions For Authors:**

- Five seeds are the bare minimum and I read in the appendix that the shades in the graphs are STD. could you perform a confidence internal analysis and show the confidence interval instead of the STD?

- The shades in Fig 6C seem too narrow to be originating from 5 different seeds, see 6A and 6B in comparison. Can you double check this detail?

**Relation To Broader Scientific Literature:**

There is a solid link with the recent literature on plasticity loss. The ideas in this paper clearly stem from recent advances in this area.

The overall objective to maintain plasticity while reducing forgetting, however, is related to the broader field of lifelong learning which is not particularly expanded upon.

**Theoretical Claims:**

The paper is largely empirical with limited or no theoretical claims.

---

### Decision · Program_Chairs · 2025-05-01

**Decision:**

Accept (poster)

**Comment:**

There is clearly some disagreement among reviewers. Reviewer Zet6 gave a detailed explanation for his negative decision and engaged with authors. However, I believe the details of Figure 1, which is mostly there to introduce the general problem, may be of less relevance than actual reported results in Figures 3 and following. Also, the lesser impact on SAC does not seem like a fatal flaw to me.

Therefore, overall I am leaning towards acceptance, under the condition that the relevant paper suggested by reviewer Zet6 [1] must be cited. I believe the method's novelty and results, such as they are, more than balance the (valid) observations of reviewer Zet6.

[1] "Reset & Distill: A Recipe for Overcoming Negative Transfer in Continual Reinforcement Learning"